# A robust cusum control chart for median absolute deviation based on trimming and winsorization

**Umair Khalil**[1]*, **Tahira Saeed Khan**[1], **Walaa Ahmad Hamdi**[2]*, **Dost Muhammad Khan**[1], **Muhammad Hamraz**[1]

**1** Department of Statistics, Abdul Wali Khan University Mardan, Mardan, Pakistan, **2** Department of Mathematics and Statistics, College of Science, University of Jeddah, Jeddah, Saudi Arabia

* umairkhalil@awkum.edu.pk (UK); whamdi@uj.edu.sa (WAH)

**Data Availability Statement:** All data are generated by simulation.

**Funding:** The author(s) received no specific funding for this work.

## Abstract

Statistical quality control is concerned with the analysis of production and manufacturing processes. Control charts are process control techniques, commonly applied to observe and control deviations. Shewhart control charts are very sensitive and used for large shifts based on the basic assumption of normality. Cumulative Sum (*CUSUM*) control charts are effective for identifying that may have special causes, such as outliers or excessive variability in subgroup means. This study uses a *CUSUM* control chart problems structure to evaluate the performance of robust dispersion parameters. We investigated the design structure features of various control charts, based on currently defined estimators and some new robust scale estimators using trimming and winsorization in different scenarios. The Median Absolute Deviation based on trimming and winsorization is introduced. The effectiveness of *CUSUM* control charts based on these estimators is evaluated in terms of average run length (*ARL*) and Standard Deviation of the Run Length (*SDRL*) using a simulation study. The results show the robustness of the *CUSUM* chart in observing small changes in magnitude for both normal and contaminated data. In general, robust estimators *MADTM* and *MADWM* based on *CUSUM* charts outperform in all environments.

## 1 Introduction

Statistical process control (*SPC*) is a method used in quality control to apply statistical techniques for monitoring and managing a system. The initiation of *SPC* occurs during the planning phase of a product or service when the relevant attributes are specified. In 1931, Shewhart introduced the concept of control charts, a pivotal technique in *SPC*. However, the effectiveness of these control charts diminishes when the assumption of normality is violated, and outliers are present in the data.

For enhanced robustness, it is desirable to have control charts that are less influenced by violations of fundamental assumptions. The selection of control charts depends on the process attribute under consideration and the type of change or shift quantity to be evaluated. Control

**Competing interests:** The authors have declared that no competing interests exist.

charts are broadly classified into two categories: memoryless control charts and memory control charts.

Memoryless control charts, often referred to as Shewhart-type control charts, are less sensitive to small and moderate parameter changes in location and dispersion. On the other hand, memory control charts, such as *CUSUM* control charts [1–3] and exponentially weighted moving average (*EWMA*) control charts [4–7], which are designed to address issues related to outliers and deviations from normality.

The *CUSUM* charts have gained popularity in quality control due to their simplicity and efficiency, initially used for monitoring mean levels of processes [8, 9]. However, their application for measuring process variability has received less attention. Hawkins suggested a robust chart for individual observations based on winsorization, while Lucas and Crosier explored methods to enhance the robustness of standard *CUSUM* charts [10–12].

The study by Lee et al. [13], proposed *CUSUM* charts for systematically correlated data, Wang et al. introduced a nonparametric *CUSUM* chart focused on the Mann-Whitney statistic, and Wang et al. [14, 15] suggested an adaptive multivariate *CUSUM* chart. Moustafa [16] introduced modified Shewhart charts for median and median absolute deviations as robust location and dispersion estimators.

Ou et al. [17, 18] conducted a comparison study on the performance of various control charts, including standard $\overline{X}$ charts, *CUSUM*, and sequential probability ratio test *SPRT* control charts, considering special situations such as trimmed and winsorized means. Wang et al. [19] introduced Trimmed and Winsorized means for transformed data based on scaled deviation, which proved to be more robust.

The Maxwell *CUSUM* control chart, proposed by Hossain et al. [20], efficiently monitors failure rates in boring processes. The *VCUSUM* chart, based on a Maxwell distribution, has been developed to detect tiny changes in a process. Castagliola et al. [21] used the *CUSUM* median chart, and Moustafa et al. [22] suggested *MTSD-TCC*, a robust control chart based on the modified trimmed standard deviation (*MTSD*) as an alternative to Tukey's control chart (*TCC)*.

This paper aims to enhance the efficiency of *CUSUM* control charts by modifying the use of dispersion parameters and comparing the efficiency of robust estimators in different environments. The investigation includes the performance of *CUSUM* control charts in uncontaminated and contaminated environments with symmetric and asymmetric variance disturbances, as well as non-normal environments, using Average Run Length (*ARL*) and Standard Deviation of the Run Length (*SDRL*).

To facilitate interpretation, the discussion will focus on the upper side of the *CUSUM* control charts, with a note that double-sided *CUSUM* control charts exhibit qualitative similarity. The remaining sections of the paper are organized as follows: Section 2 describes dispersion estimators, Section 3 presents proposed estimators, and Section 4 outlines the proposed *CUSUM* control chart (Fig 1) with different robust dispersion estimators based on trimmed and winsorization. Finally, major conclusions are summarized in the closing section.

## 2 Description of process dispersion estimators

Let $\vartheta$ be the parameter of the process dispersion that needs to be controlled by control charts and $\hat{\vartheta}$ be the estimator based on a sample of size $n$. For $\hat{\vartheta}$ there are several choices. David [23] gives a clear description of standard deviation estimators. Typical estimators are the average of the sample standard deviations, pooled sample standard deviation, and average of sample ranges. Mahmoud et al. [24] investigated the relative ability of estimators for different $k$ samples of size $n$. Schoonhoven et al. [25] considered various estimators of the population standard

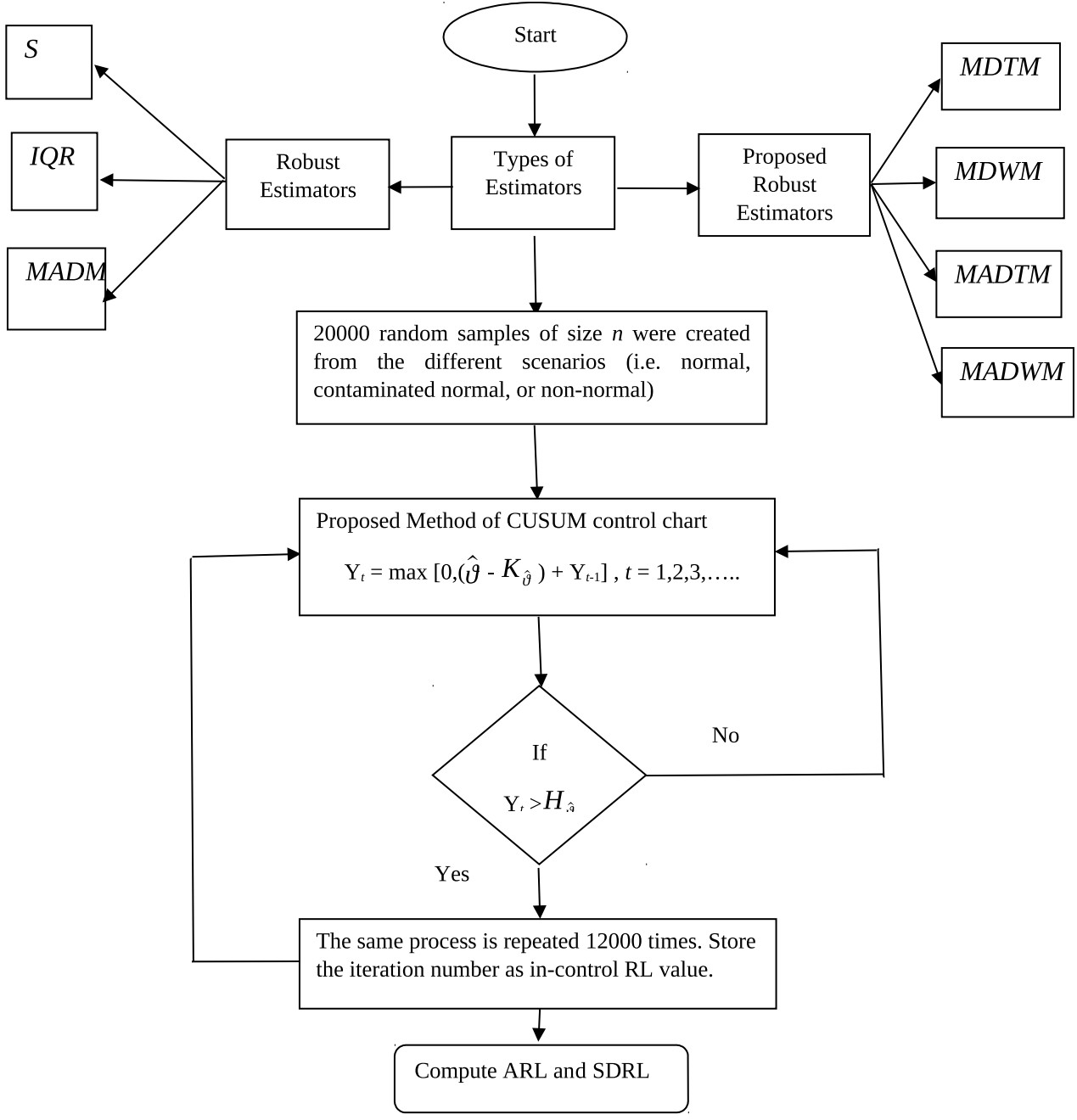

**Fig 1. Flowchart procedure for the proposed method of *CUSUM* control chart.**

deviation and presented a detailed overview of their efficiency and use for different stages in the control chart.

The following estimators are used in this paper, which is described:

The first estimator of $\vartheta$ is the sample standard deviation $S$ defined as:

$$S = \sqrt{\frac{\sum_{i=1}^{n}(Y_i - \overline{Y})2}{n-1}}, \qquad (1)$$

where $Y_i$ indicates the $i^{\text{th}}$ observation of sample size $n$ and $\overline{Y}$ indicate the sample mean. In a normally distributed environment, the sample standard deviation $S$ is the most effective estimator but is strongly influenced by outliers. The sample standard deviation breakdown point (the ratio of outlying observations that an estimator can deal with) is zero.

The sample interquartile range ($IQR$) is the next estimator based on $CUSUM\text{-}\hat{\vartheta}$ charts which are defined by

$$IQR = \frac{Q_3 - Q_1}{1.34898}, \tag{2}$$

where $Q_1$ and $Q_3$ are the first and the third quartiles of the sample, respectively. The sample interquartile range is more stable than the sample standard deviation [26]. The breakdown point of $IQR$ is 25%.

The median absolute deviation from the sample median ($MADM$) is a very robust dispersion estimator rather than the sample standard deviation. It calculates the differences of the data from the median of the sample. The $MADM$ is defined as:

$$MADM = Median \; Y_i - \tilde{Y} \tag{3}$$

and

$$\hat{\sigma} = 1.4826 \; MADM$$

where the sample median is $\tilde{Y}$. For the parameter of interest, the constant 1.4826 is required to make the estimator compatible. In case of normal distribution, $\sigma$ normal parameter is required to set 1.4826. (In the case of an unbiased estimator of σ, we need to set this constant to 1.4826 if a random sample is taken from a normal distribution. Median Absolute Deviation is 1.4826 times the Median of Absolute Differences of Individual Values of a Dataset from the Median of the Dataset) (Supporting Data).

## 3 Proposed estimators based on trimmed winsorization

The trimmed mean is a relatively robust estimate of the centre, which decreases the effect of outliers or large tails by eliminating the observations at the distribution. Let $Y_1, Y_2, \cdots, Y_n$, represents observations on a variable from a random sample of size $n$. We begin by arranging the $Y$ values from smallest to largest, $Y_1 \leq Y_2 \leq \cdots \leq Y_n$, and measuring the number of trimmings required. The symmetrically trimmed sample of $k$-times is obtained by removing the $k$-smallest and $k$-largest values. In this case, $k = (\alpha n)$ is the largest integer, and trimming is done for $\alpha$ % ($0 \leq \alpha \leq \frac{1}{2}$) of $n$. The trimmed mean $\overline{Y}_T$ is defined as:

$$\overline{Y}_T = \frac{\sum_{i=k+1}^{n-k} Y_i}{n - 2k}. \tag{4}$$

The breakdown point is calculated by the number of trimmings thus $BDP = \alpha$. A basic rule of thumb is to deduct from each tail of the distribution 10% of the observations (i.e., set $\alpha = 0.2$). Mean deviation from trimmed mean $MDTM$ is defined as:

$$MDTM = \frac{\sum_{i=1}^{n} |Y_i - \overline{Y}_T|}{n}. \tag{5}$$

The next proposed estimator is the median of the absolute deviations from the trimmed mean, $MADTM$ is defined as:

$$MADTM = 1.4826 \, Median \, |Y_i - \overline{Y}_T|. \tag{6}$$

The method of substituting a given number of extreme values with having small values has become known as winsorizing data or winsorization. Let $Y_1, Y_2, \cdots, Y_n$, represents observations on a variable from a random sample of size $n$. The data of $Y$ values are sorted from smallest to largest, i.e $Y_1 \leq Y_2 \leq \cdots \leq Y_n$, and the smallest $k$ values are replaced with the smallest $(k+1)^{st}$ values. The same process is valid for the largest values, substituting the largest $k$ values with the largest $(k+1)^{st}$ value. The mean is known as the winsorized mean in this new set of numbers. The winsorized mean is a robust, unbiased approximation of the population mean if the data are from a symmetric population. The $k$ times winsorized mean $\overline{Y}_W$ is defined as:

$$\overline{Y}_W = \frac{1}{n} \left( (k+1) \, Y_{(k+1)} + \sum_{i=k+2}^{n-k-1} Y_i + (k+1) Y_{(n-k)} \right). \tag{7}$$

The mean deviation from the winsorized mean $MDWM$ is

$$MDWM = \frac{\sum_{i=1}^{n} |Y_i - \overline{Y}_w|}{n}. \tag{8}$$

The next proposed estimator is the median of the absolute deviations from the winsorized mean, $MADWM$ is defined as:

$$MADWM = 1.4826 \, Median \, |Y_i - \overline{Y}_w|. \tag{9}$$

For comparison and to determine the precision of the dispersion robust estimators used in this analysis, the standardized variances of the estimators as proposed by Rousseeuw and Croux [27] and relative efficiencies of the estimators as suggested by Abbasi and Miller [28] are calculated.

The dispersion estimator $\hat{\vartheta}$ of standardized variance $(SV_{\hat{\vartheta}})$ is measured as:

$$SV_{\hat{\vartheta}} = \frac{n \, var\left(\hat{\vartheta}\right)}{\left[ E\left(\hat{\vartheta}\right) \right]^2} \tag{10}$$

To obtain a normal measure of the precision of a scale estimator the denominator of $SV_{\hat{\vartheta}}$ is necessary [29]. The estimator's relative efficiency $(RE_{\hat{\vartheta}})$ is calculated as:

$$RE_{\hat{\vartheta}} = \frac{min(SV_{\hat{\vartheta}})}{SV_{\hat{\vartheta}}} \tag{11}$$

First, the $SV_{\hat{\vartheta}}$ and $RE_{\hat{\vartheta}}$ values for all robust estimators are computed and compared. A simulation study is used to check the performance of robust estimators based on $CUSUM$-$\hat{\vartheta}$ charts. The simulation is run 20,000 times, $SV_{\hat{\vartheta}}$ and $RE_{\hat{\vartheta}}$ are determined by samples of size $n = 5, 6 \, and \, 9$ based on the following conditions: uncontaminated normal, contaminated normal, gamma, and logistic scenarios. The $SV_{\hat{\vartheta}}$ and $RE_{\hat{\vartheta}}$ results are listed in Tables 1 and 2 under different scenarios of dispersion estimators based on $CUSUM$-$\hat{\vartheta}$ charts. Results given in the tables of $SV_{\hat{\vartheta}}$ and $RE_{\hat{\vartheta}}$ suggested that, under the uncontaminated normal scenario the $S$ has the largest $SV_{\hat{\vartheta}}$ but smallest $SV_{\hat{\vartheta}}$ under Logistic distribution with sample size $n = 9$. The smallest $SV_{\hat{\vartheta}}$ of the dispersion estimator is $MADWM$(with 25% winsorizing) for a small sample size

**Table 1. Standardized variance of robust estimators in different scenarios.**

| Scenarios | Sample Size | Estimators | | | | | | | | | | | | | | |
|---|---|---|---|---|---|---|---|---|---|---|---|---|---|---|---|---|
| | | S | IQR | MADM | MDTM | | | MDWM | | | MADTM | | | MADWM | | |
| | | | | | 10% | 20% | 25% | 10% | 20% | 25% | 10% | 20% | 25% | 10% | 20% | 25% |
| N(0,1) | 5 | 0.1674 | 0.1980 | 0.1002 | 0.4353 | 0.4452 | 0.4342 | 0.3975 | 0.4437 | 0.3946 | 0.1079 | 0.0975 | 0.1035 | 0.1058 | 0.0909 | **0.0887** |
| | 6 | 1.0447 | 2.5014 | 0.1102 | 0.5042 | 0.4777 | 0.4783 | 0.4673 | 0.4708 | 0.4865 | 0.1070 | **0.0967** | 0.1085 | 0.0993 | 0.1027 | 0.1095 |
| | 9 | *2.7168* | 2.1078 | 0.1309 | 0.6611 | 0.6352 | 0.6511 | 0.7020 | 0.6589 | 0.6473 | 0.1173 | 0.1167 | 0.1935 | **0.1057** | 0.1096 | 0.1101 |
| 1% C Normal | 5 | 0.1389 | 0.1825 | 0.1144 | 0.4502 | 0.4462 | 0.4565 | 0.3903 | 0.4277 | 0.3924 | 0.0968 | 0.0976 | **0.0818** | 0.0977 | 0.0896 | 0.0872 |
| | 6 | 0.7342 | 1.1719 | 0.1088 | 0.4891 | 0.4807 | 0.5041 | 0.4805 | 0.4275 | 0.4763 | 0.1121 | **0.0931** | 0.1087 | 0.1090 | 0.1000 | 0.1045 |
| | 9 | 1.3784 | 1.4649 | 0.1100 | 0.6875 | 0.6415 | 0.6829 | 0.6288 | 0.7332 | 0.6654 | **0.1041** | 0.1183 | 0.2980 | 0.1160 | 0.1238 | 0.1199 |
| 5% C Normal | 5 | 0.1474 | 0.2191 | 0.1045 | 0.4401 | 0.4247 | 0.4461 | 0.4279 | 0.4143 | 0.4288 | 0.0993 | 0.0972 | 0.1005 | **0.0891** | 0.0962 | 0.1000 |
| | 6 | 0.9788 | 1.1434 | 0.1090 | 0.4794 | 0.4926 | 0.5199 | 0.4450 | 0.5035 | 0.4727 | **0.0970** | 0.1067 | 0.1022 | 0.1013 | 0.1028 | 0.1009 |
| | 9 | 1.3789 | 1.2815 | 0.1469 | 0.6615 | 0.7075 | 0.6823 | 0.6805 | 0.6633 | 0.6798 | 0.1135 | 0.1143 | 0.1906 | **0.0972** | 0.1123 | 0.1124 |
| 10% C Normal | 5 | 0.2529 | 0.2816 | 0.1070 | 0.4419 | 0.4030 | 0.4574 | 0.4472 | 0.4123 | 0.4285 | 0.1042 | 0.0923 | 0.0937 | 0.0954 | **0.0919** | 0.1056 |
| | 6 | 0.8730 | 1.0491 | 0.1638 | 0.5086 | 0.5238 | 0.4696 | 0.4631 | 0.4656 | 0.4478 | **0.0945** | 0.1043 | 0.1083 | 0.1011 | 0.1001 | 0.1199 |
| | 9 | 1.3666 | 1.4432 | 0.1125 | 0.6737 | 0.6752 | 0.6892 | 0.6773 | 0.6556 | 0.7096 | 0.1762 | 0.1119 | 0.2841 | 0.1129 | 0.1154 | **0.1073** |
| 15% C Normal | 5 | 0.2610 | 0.4477 | 0.1216 | 0.4366 | 0.4449 | 0.4325 | 0.4265 | 0.4020 | 0.3716 | 0.1012 | 0.0969 | 0.0942 | 0.0993 | 0.0935 | **0.0806** |
| | 6 | 0.9952 | 1.2509 | 0.1451 | 0.4697 | 0.4739 | 0.4640 | 0.4394 | 0.4573 | 0.4619 | 0.1017 | 0.1057 | **0.0946** | 0.0992 | 0.0994 | 0.1081 |
| | 9 | 1.4877 | 1.2847 | 0.3021 | 0.6977 | 0.6735 | 0.6695 | 0.6743 | 0.6678 | 0.6442 | 0.1260 | 0.1593 | 0.3285 | **0.1009** | 0.1178 | 0.1016 |
| Logistic (2,1) | 5 | 0.3262 | **0.2711** | 0.4232 | 0.7235 | 0.7542 | 0.7741 | 0.7307 | 0.7614 | 0.7386 | 0.3777 | 0.3777 | 0.3461 | 0.3743 | 0.3531 | 0.3843 |
| | 6 | 0.3377 | **0.2861** | 0.3375 | 0.8166 | 0.8683 | 0.8671 | 0.8096 | 0.7349 | 0.8170 | 0.3760 | 0.3854 | 0.4355 | 0.4093 | 0.3957 | 0.4226 |
| | 9 | **0.3690** | 0.3772 | 0.4929 | 0.8691 | 0.8812 | 0.9038 | 0.8473 | 0.8356 | 0.8932 | 0.4246 | 0.4600 | 0.4590 | 0.4910 | 0.4365 | 0.4426 |
| G(2,1) | 5 | 0.2765 | 0.2836 | 0.2837 | 0.8459 | 0.7697 | 0.7083 | 0.7652 | 0.7778 | 0.7415 | 0.2636 | 0.2565 | **0.2411** | 0.2817 | 0.2951 | 0.3109 |
| | 6 | 0.4935 | **0.2399** | 0.2406 | 0.8754 | 0.8281 | 0.7790 | 0.7885 | 0.8308 | 0.8127 | 0.2922 | 0.2409 | 0.2587 | 0.2993 | 0.2839 | 0.3017 |
| | 9 | 0.3122 | 0.3063 | 0.2689 | 1.0886 | 0.9636 | 0.9120 | 0.9428 | 0.9596 | 0.9356 | 0.3619 | 0.2789 | **0.2393** | 0.3364 | 0.3026 | 0.3287 |

$n = 5$ under 15% contaminated normal scenario. The efficiency of other dispersion estimators lies between these two estimators *MADWM* (with 10%, 20%, and 25% winsorizing) and *S* estimators. The *IQR* has the smallest $SV_{\hat{\vartheta}}$ under logistic distribution and gamma distribution. Under 5%, 10%, and 15% symmetric variance contaminated normal scenario the *MADTM*(at 10% and 25% trimming) with sample size $n = 6$ has the smallest $SV_{\hat{\vartheta}}$. Under 5% symmetric variance contaminated normal scenario the *MADWM*(at 10% winsorizing) with sample size $n = 5$ *and* 9, 10% contaminated Normal scenario the *MADWM*(at 20% and 25% winsorizing) with sample size $n = 5$ *and* 9 and 15% contaminated Normal scenario the *MADWM*(at 25% winsorizing) with sample size $n = 5$ has the smallest $SV_{\hat{\vartheta}}$. The *MADTM*(at 10%, 20%, and 25% trimming) obtains the smallest $SV_{\hat{\vartheta}}$ for small 1% contamination with sample sizes $n = 5,6$ *and* 9. For Gamma distribution, the *MADTM*(at 25% trimming) with sample size $n = 5$ has the lowest value of $SV_{\hat{\vartheta}}$. Particularly for the non-normal scenarios, the *MADTM*(at 10%, 20%, and 25% trimming) and *MADWM* (at 10%, 20%, and 25% winsorizing) performance is best in contrast to the rest of all other estimators. The dispersion estimators *MADM*, *MDTM*(at 10%, 20%, and 25% trimming) and *MDWM* (at 10%, 20%, and 25% winsorizing) are highly affected by contaminations and non-normal environments. It shows that proposed estimators *MADTM*(at 10%, 20%, and 25% trimming) and *MADWM* (at 10%, 20%, and 25% winsorizing) performance is more efficient than other estimators.

**Table 2. Relative efficiency of robust estimators in different scenarios.**

| Scenarios | Sample Size | Estimators | | | MDTM | | | MDWM | | | MADTM | | | MADWM | | |
|---|---|---|---|---|---|---|---|---|---|---|---|---|---|---|---|---|
| | | S | IQR | MADM | 10% | 20% | 25% | 10% | 20% | 25% | 10% | 20% | 25% | 10% | 20% | 25% |
| N(0,1) | 5 | 52.9778 | 44.7799 | 88.5486 | 20.3734 | 19.9199 | 20.4256 | 22.3081 | 19.9874 | 22.4754 | 82.2054 | 90.9676 | 85.6997 | 83.7968 | 97.5567 | **100.0000** |
| | 6 | 9.2522 | 3.8641 | 87.6855 | 19.1712 | 20.2348 | 20.2089 | 20.6818 | 20.5310 | 19.8688 | 90.3443 | **100.0000** | 89.0612 | 97.3637 | 94.1405 | 88.2453 |
| | 9 | 3.8894 | 5.0134 | 80.7276 | 15.9828 | 16.6362 | 16.2293 | 15.0517 | 16.0373 | 16.3246 | 90.1229 | 90.7302 | 54.6130 | **100.0000** | 96.4304 | 95.9597 |
| 1% C Normal | 5 | 58.8533 | 44.7843 | 71.4207 | 18.1541 | 18.3175 | 17.9051 | 20.9418 | 19.1092 | 20.8284 | 84.4343 | 83.8320 | **100.0000** | 83.6248 | 91.1988 | 93.7540 |
| | 6 | 12.6802 | 7.9442 | 85.5467 | 19.0338 | 19.3681 | 18.4699 | 19.3752 | 21.7768 | 19.5455 | 83.0194 | **100.0000** | 85.6301 | 85.4489 | 93.1243 | 89.1178 |
| | 9 | 7.5537 | 7.1077 | 94.6199 | 15.1444 | 16.2316 | 15.2468 | 16.5583 | 14.2007 | 15.6471 | **100.0000** | 88.0188 | 34.9397 | 89.7644 | 84.1227 | 86.8651 |
| 5% C Normal | 5 | 60.4040 | 40.6387 | 85.2277 | 20.2345 | 20.9677 | 19.9612 | 20.8120 | 21.4954 | 20.7696 | 89.6881 | 91.6355 | 88.6015 | **100.0000** | 92.5655 | 89.0993 |
| | 6 | 9.9101 | 8.4836 | 88.9667 | 20.2329 | 19.6920 | 18.6578 | 21.7973 | 19.2657 | 20.5227 | **100.0000** | 90.9329 | 94.9487 | 95.7760 | 94.3641 | 96.1801 |
| | 9 | 7.0467 | 7.5822 | 66.1637 | 14.6889 | 13.7338 | 14.2420 | 14.2798 | 14.6483 | 14.2932 | 85.5802 | 84.9928 | 50.9921 | **100.0000** | 86.5554 | 86.4892 |
| 10% C Normal | 5 | 36.3268 | 32.6185 | 85.8517 | 20.7892 | 22.7971 | 20.0842 | 20.5428 | 22.2809 | 21.4379 | 88.1495 | 99.4880 | 98.0702 | 96.2632 | **100.0000** | 86.9553 |
| | 6 | 10.8189 | 9.0034 | 57.6791 | 18.5720 | 18.0313 | 20.1136 | 20.3960 | 20.2879 | 21.0946 | **100.0000** | 90.5269 | 87.1829 | 93.4451 | 94.3943 | 78.7973 |
| | 9 | 7.85261 | 7.4356 | 95.3925 | 15.9288 | 15.8929 | 15.5704 | 15.8451 | 16.3679 | 15.1241 | 60.8929 | 95.9246 | 37.7703 | 95.0453 | 93.0317 | **100.0000** |
| 15% C Normal | 5 | 30.8816 | 18.0065 | 66.2822 | 18.4647 | 18.1176 | 18.6400 | 18.9013 | 20.0524 | 21.6956 | 79.6460 | 83.2078 | 85.5553 | 81.2166 | 86.2582 | **100.0000** |
| | 6 | 9.5069 | 7.5635 | 65.2244 | 20.1418 | 19.9624 | 20.3895 | 21.5335 | 20.6899 | 20.4818 | 93.0301 | 89.4902 | **100.0000** | 95.3999 | 95.1590 | 87.5304 |
| | 9 | 6.7800 | 7.8511 | 33.3895 | 14.4558 | 14.9758 | 15.0644 | 14.9578 | 15.1031 | 15.6562 | 80.0773 | 63.3058 | 30.7088 | **100.0000** | 85.6356 | 99.3009 |
| Logistic (2,1) | 5 | 83.1014 | **100.0000** | 64.0499 | 37.4628 | 35.9374 | 35.0139 | 37.0958 | 35.5969 | 36.6963 | 71.7574 | 71.7719 | 78.3243 | 72.4251 | 76.7547 | 70.5242 |
| | 6 | 84.7201 | **100.0000** | 84.7856 | 35.0381 | 32.9504 | 32.9969 | 35.3406 | 38.9335 | 35.0209 | 76.0934 | 74.2472 | 65.7002 | 69.8998 | 72.3102 | 67.7051 |
| | 9 | **100.0000** | 97.8044 | 74.8373 | 42.4437 | 41.8624 | 40.8142 | 43.5377 | 44.1476 | 41.2998 | 86.8909 | 80.2032 | 80.3662 | 75.1272 | 84.5073 | 83.3406 |
| G(2,1) | 5 | 87.1982 | 85.0237 | 85.0005 | 28.5068 | 31.3281 | 34.0426 | 31.5097 | 31.0021 | 32.5199 | 91.4852 | 94.0216 | **100.0000** | 85.6093 | 81.7036 | 77.5682 |
| | 6 | 48.6147 | **100.0000** | 99.7006 | 27.4067 | 28.9715 | 30.7995 | 30.4287 | 28.8783 | 29.5229 | 82.0988 | 99.5942 | 92.7296 | 80.1625 | 84.5134 | 79.5295 |
| | 9 | 76.6741 | 78.1548 | 89.0142 | 21.9879 | 24.8383 | 26.2441 | 25.3881 | 24.9425 | 25.5820 | 66.1427 | 85.8292 | **100.0000** | 71.1591 | 79.1048 | 72.8132 |

## 4 The proposed method of CUSUM charts for different robust dispersion estimators

For the *CUSUM* procedures, identify a way to increase the dispersion process parameter $\vartheta$. Let $\hat{\vartheta}$ be an estimator from Section 2 of the dispersion process parameter $\vartheta$ from a random sample of size $n$ that is taken.at regular intervals from a continuous production process. The *CUSUM-$\hat{\vartheta}$* chart is defined as:

$$Y_t = \max[0, (\hat{\vartheta} - K_{\hat{\vartheta}}) + Y_{t-1}], \ t = 1, 2, 3, \ldots \tag{12}$$

According to Tuprah and Ncube [30] where $Y_0 = 0$ and the reference value of the scheme is $K_{\hat{\vartheta}}$. $Y_t$ is plotted against the sample number $t$. The process is assumed to be out of reach if $Y_t > H_{\hat{\vartheta}}$ (where $H_{\hat{\vartheta}}$ defines the decision interval) for any value of $t$ and it is concluded that the dispersion of the process has increased. The procedure of average run length is the expected value of the run length of the process and the random variable run length for the sample number at which $Y_t > H_{\hat{\vartheta}}$. The $H_{\hat{\vartheta}}$ values are selected such that changes in the dispersion of process parameters are easily identified. When the system is in control in all the scenarios considered in this analysis, $H_{\hat{\vartheta}}$ values are selected for a fixed value of *ARL* along with the $K_{\hat{\vartheta}}$ value and is denoted by $ARL_0$. $ARL_1$ stands for the out-of-control *ARL*, which is predicted to be as small as possible. The reference value $K_{\hat{\vartheta}}$ is based on Tuprah and Ncube [30], Ewan and Kemp [31], and E.S. Page [32], so the value $K_{\hat{\vartheta}}$ is taken as half of the expected values of $\hat{\vartheta}$ given $\vartheta_0 = 1$ and the expected values of $\hat{\vartheta}$ given $\vartheta_1 = 1.4$, where $\vartheta_0$ is the target value and $\vartheta_1$ is the value of dispersion process that needs to be easily detected. E.S. Page [32] in Table 1, presented the reference values for noticing a change (that is $\vartheta_1 = 1.40$ to $\vartheta_1 = 2.23$) easily in the dispersion of the process using the sample range.

$$K_{\hat{\vartheta}} = \frac{\left[ E\left(\hat{\vartheta} \mid \vartheta_0\right) + E\left(\hat{\vartheta} \mid \vartheta_1\right) \right]}{2}. \tag{13}$$

Accordingly, for $K_{\hat{\vartheta}}$ it is difficult to find the value of $E\left(\hat{\vartheta} \mid .\right)$ analytically. For this purpose, simulation is used, from normal distribution random samples are generated with mean $\vartheta_0 = 1$ respectively, $\vartheta_1 = 1.40$, and variance equal to one and it calculates the said expected value.

The results of *CUSUM-$\hat{\vartheta}$* charts are obtained in the following scenarios based on Tatum [33] and Schoonhoven et al. [25].

1. A model in which all observations are from N(0,1) (i.e., uncontaminated scenario).

2. A symmetric variance disturbances model, in which each observation has a 99% probability from the distribution N(0,1) and a 1% probability from N(0,9).

3. A model of asymmetric variance disturbances, in which each observation is taken from an N(0,1) and has a 1% probability of adding a multiple of a $\chi_1^2$ variable to it, with a multiplier equal to 4.

4. We consider two situations to examine the impact of non-normal distributions: the first contains disturbing the kurtosis, and the second involves disturbing the symmetry distribution. We use Logistic distribution Logistic (2,1) for the disturbance of kurtosis and the gamma distribution for the disturbance in symmetry G(2,1).

Tables 3 and 4 show the values of $K_{\hat{\vartheta}}$ and $H_{\hat{\vartheta}}$ for different robust estimators based on CUSUM-$\hat{\vartheta}$ charts under different scenarios (i.e. normal and non-normal).

**Table 3.** $K_{\hat{\vartheta}}$ Values for *CUSUM*-$\hat{\vartheta}$ charts in different scenarios with $ARL_O = 500$.

| Estimators | α | Scenarios | | | |
|---|---|---|---|---|---|
| | | N(0,1) | | G(2,1) | Logistic (2,1) |
| | | n = 5 | n = 9 | n = 5 | n = 5 |
| S | | 1.13 | 1.13 | 2.70 | 2.91 |
| IQR | | 1.47 | 1.34 | 2.34 | 2.72 |
| MADM | | 0.98 | 1.09 | 2.61 | 3.10 |
| MDTM | 10% | 1.12 | 1.07 | 1.73 | 2.11 |
| | 20% | 1.28 | 1.14 | 1.58 | 1.98 |
| | 25% | 1.28 | 1.11 | 1.59 | 1.99 |
| MDWM | 10% | 1.30 | 1.13 | 1.83 | 1.99 |
| | 20% | 1.507 | 1.137 | 1.76 | 2.10 |
| | 25% | 1.469 | 1.375 | 1.78 | 2.12 |
| MADTM | 10% | 1.25 | 1.469 | 2.65 | 3.20 |
| | 20% | 1.480 | 1.36 | 2.51 | 2.63 |
| | 25% | 1.483 | 1.327 | 2.56 | 2.64 |
| MADWM | 10% | 1.357 | 1.35 | 2.54 | 2.81 |
| | 20% | 2.31 | 1.35 | 2.55 | 3.30 |
| | 25% | 2.30 | 1.35 | 2.53 | 2.91 |

In different scenarios (normal and non-normal) $H_{\hat{\vartheta}}$ values are searched by selecting random samples separately from the environments described until the value of $H_{\hat{\vartheta}}$ is obtained in each case. An iterative method is used to modify the desired *ARL* as well as the $K_{\hat{\vartheta}}$ reference value. Table 4 is given with $ARL_0 = 500$ and the values of $H_{\hat{\vartheta}}$. Similarly, alternative values of $H_{\hat{\vartheta}}$

**Table 4.** $H_{\hat{\vartheta}}$ Values for *CUSUM*-$\hat{\vartheta}$ charts in different scenarios with $ARL_O = 500$.

| Estimators | α | Scenarios | | | |
|---|---|---|---|---|---|
| | | N(0,1) | | G(2,1) | Logistic (2,1) |
| | | n = 5 | n = 9 | n = 5 | n = 5 |
| S | | 1.531 | 0.916 | 1.29 | 1.58 |
| IQR | | 0.803 | 0.763 | 1.31 | 1.51 |
| MADM | | 3.13 | 1.852 | 1.13 | 1.96 |
| MDTM | 10% | 0.511 | 0.32 | 1.31 | 1.26 |
| | 20% | 0.32 | 0.27 | 1.21 | 1.30 |
| | 25% | 0.32 | 0.30 | 1.20 | 1.29 |
| MDWM | 10% | 0.30 | 0.29 | 1.10 | 1.39 |
| | 20% | 0.09 | 0.28 | 1.12 | 1.21 |
| | 25% | 0.127 | 0.03 | 1.07 | 1.18 |
| MADTM | 10% | 1.867 | 0.853 | 1.51 | 1.75 |
| | 20% | 1.126 | 1.007 | 1.29 | 2.17 |
| | 25% | 1.123 | 1.0001 | 1.24 | 2.16 |
| MADWM | 10% | 1.449 | 1.09 | 1.50 | 1.99 |
| | 20% | 0.007 | 1.09 | 1.40 | 1.31 |
| | 25% | 0.006 | 1.10 | 1.39 | 1.73 |

can be found for other values of $ARL_0$. Since the $ARL_0$ of the $CUSUM$-$\hat{\vartheta}$ chart's results are prone to these values, the $K_{\hat{\vartheta}}$ and $H_{\hat{\vartheta}}$ values must be carefully selected.

## 5 Evaluation of CUSUM—$\hat{\vartheta}$ charts performance

The $ARL$ is used as simulation method to evaluate the performance of the suggested CUSUM -$\hat{\vartheta}$ charts. The $ARL$ of in-control and out-of-control systems is calculated using the monte carlo simulation. The descriptions of the simulation are: 20000 random samples of size $n$ were created from the different scenarios (i.e. normal, contaminated normal, or non-normal) and the dispersion estimators concerned with some recent estimators (i.e. $S$, $IQR$, and $MADM$) as well as some suggested robust estimators (i.e. $MDTM$, $MDWM$, $MADTM$, and $MADWM$) based on trimming and winsorization at (10%, 20%, and 25%) are measured. Tables 3 and 4 are used to generate the corresponding limits of the control chart. It is noted that the sample number at which statistic $Y_t$ lies beyond the control limits, this sample number is known as run-length, and it is a random variable. To determine the run length distribution, the same process is repeated 12000 times. The $ARL$ represents the average of the run length distribution and SDRL represents the standard deviation of the run length distribution. To determine the run lengths a code has been built in the $R$ language.

### 5.1 Results and discussions

The $ARL_1$ and $SDRL_1$ are used in different environments to evaluate the performance and efficiency of the $CUSUM$-$\hat{\vartheta}$ charts. In terms of $\vartheta$ (i.e $\delta\vartheta$) we have identified shifts which specify that the shifted dispersion parameter $\hat{\vartheta}$ is defined as $\hat{\vartheta} = \delta\vartheta$. Here $\delta = 1$ indicates that there is no shift in $\vartheta$ and the dispersion of the process is constant, and $\delta > 1$ indicates that the process $\vartheta$ has increased. $ARL_1$ increases when the process shift decreases. $SDRL$ decreases as the size of the process shift increases. It depends on the size of the shift. When the process is in control, the $ARL$ and $SDRL$ process to be close to its targeted value namely 500 In all environments, robust $MADTM$ and $MADWM$ estimators based on $CUSUM$ charts work well.

**5.1.1 Uncontaminated environment.** All observations are normally distributed in an uncontaminated environment $N(0,1)$. This environment is the fundamental assumption of the design structure of each chart. This provides a conceptual framework for comparing the various types of control charts and the suggested $CUSUM$-$\hat{\vartheta}$ chart. Table 5 shows the results of $ARL$.

A large value of $ARL$ is desired when the process is stable or in control. In Table 5 the bold letter shows the highest score of ARL of robust estimators at different levels of trimming and winsorization with sample sizes of $n = 5$ $and$ $n = 9$. It can be seen that the Standard deviation $S$ based on the $CUSUM$-$\hat{\vartheta}$ chart of sample size $n = 5$ has the best performer as compared to $IQR$, $MADM$ highlighted values in Table 5. The proposed estimator $MDTM$ (at 10%, 20%, and 25% trimming) performance is best for both sample sizes ($n = 5$ $and$ 9) as compared to $S$, $IQR$, $MADM$. For both sample sizes $n = 5$ $and$ $n = 9$ when the shift $\delta > 1.25$ the $MADTM$ (at 10%, 20%, and 25% trimming) and the $MADWM$ (at 10%, 20% and 25% winsorizing) performs better as compared to the $S$, $IQR$, and $MADM$. The ARL of proposed estimator the $MADTM$(at 10%, 20%, and 25% trimming) and the $MADWM$ (at 10%, 20%, and 25% winsorizing) are large than all other estimators for both sample size ($n = 5$ $and$ 9). It shows that the performance of both proposed estimators is best.

To further clarify the distribution of run lengths in an environment of the uncontaminated case, the $SDRL$ of the $CUSUM$-$\hat{\vartheta}$ charts is often recorded to measure the performance of run-length as proposed by Antzoulakos and Rakitzisis [34]. Table 6 shows the details. The $SDRL$ process is to be close to its targeted value namely 500 when the process is in control. Table 6

**Table 5.** *ARL* values of robust estimators based on *CUSUM*-$\hat{\vartheta}$ charts in uncontaminated environment N(0,1) when $ARL_O$ = 500.

| n | Estimator | α | δ | | | | | | | | |
|---|-----------|---|------|------|------|------|------|------|------|------|------|
| | | | 1.00 | 1.25 | 1.50 | 1.75 | 2.00 | 2.25 | 2.50 | 2.75 | 3.00 |
| 5 | S | | 503.4060 | 16.8001 | 6.1394 | 3.8160 | 2.8438 | 2.2991 | 1.9568 | 1.7290 | 1.5719 |
| | IQR | | 391.8081 | 47.4739 | 15.2517 | 7.7748 | 5.0088 | 3.7081 | 2.9763 | 2.5179 | 2.2192 |
| | MADM | | 403.6135 | 30.7496 | 12.3863 | 7.7116 | 5.7149 | 4.5728 | 3.8713 | 3.3701 | 3.0026 |
| | MDTM | 10% | 428.9983 | 27.8560 | 7.5133 | 3.8089 | 2.5478 | 1.9859 | 1.6593 | 1.4757 | 1.3518 |
| | | 20% | 519.4121 | 38.7963 | 9.9147 | 4.6178 | 2.9005 | 2.1521 | 1.7576 | 1.5249 | 1.3898 |
| | | 25% | 519.4121 | 38.7963 | 9.9147 | 4.6178 | 2.9005 | 2.1521 | 1.7576 | 1.5249 | 1.3898 |
| | MDWM | 10% | 427.0983 | 34.5797 | 9.1513 | 4.3481 | 2.7689 | 2.0670 | 1.6968 | 1.4928 | 1.3597 |
| | | 20% | 438.3873 | 37.7240 | 10.2317 | 4.7598 | 2.9614 | 2.1625 | 1.7562 | 1.5176 | 1.3821 |
| | | 25% | 435.7911 | 37.5660 | 10.1791 | 4.7564 | 2.9510 | 2.1648 | 1.7558 | 1.5182 | 1.3827 |
| | MADTM | 10% | 532.2948 | 31.8774 | 10.0487 | 5.6875 | 4.0122 | 3.1464 | 2.6268 | 2.2910 | 2.0493 |
| | | **20%** | **567.9308** | **53.3600** | **15.5805** | **7.6963** | **4.9683** | **3.6723** | **2.9537** | **2.5095** | **2.2148** |
| | | **25%** | **571.8143** | **53.5881** | **15.6617** | **7.7198** | **4.9765** | **3.6767** | **2.9567** | **2.5116** | **2.2168** |
| | MADWM | **10%** | **571.9853** | **41.2259** | **11.8382** | **6.1631** | **4.1313** | **3.1582** | **2.6052** | **2.2385** | **1.9993** |
| | | 20% | 230.4432 | 35.6443 | 12.6663 | 6.6794 | 4.3502 | 3.1983 | 2.5568 | 2.1839 | 1.9123 |
| | | 25% | 232.5157 | 36.3084 | 12.9369 | 6.8188 | 4.4543 | 3.2653 | 2.6088 | 2.2283 | 1.9572 |
| 9 | S | | 438.9139 | 9.43492 | 3.6087 | 2.3400 | 1.7908 | 1.4868 | 1.3072 | 1.1884 | 1.1191 |
| | IQR | | 462.6282 | 32.4068 | 8.9484 | 4.4623 | 2.9603 | 2.2811 | 1.9081 | 1.6613 | 1.5004 |
| | MADM | | 405.0183 | 19.7510 | 7.4455 | 4.5978 | 3.4036 | 2.7444 | 2.345 | 2.055 | 1.8568 |
| | MDTM | 10% | 546.9975 | 20.9249 | 4.7174 | 2.3912 | 1.6813 | 1.3623 | 1.2028 | 1.1168 | 1.0713 |
| | | 20% | 549.9122 | 22.2550 | 4.9564 | 2.4636 | 1.7103 | 1.3707 | 1.2082 | 1.1222 | 1.0726 |
| | | **25%** | **576.2238** | 22.1878 | 4.9098 | 2.4623 | 1.7155 | 1.3782 | 1.2173 | 1.1264 | 1.0771 |
| | MDWM | 10% | 559.9072 | 21.7714 | 4.8333 | 2.4234 | 1.698 | 1.3673 | 1.207 | 1.1194 | 1.0723 |
| | | 20% | 546.7148 | 21.8098 | 4.8680 | 2.4244 | 1.6984 | 1.3663 | 1.2067 | 1.1198 | 1.0718 |
| | | 25% | 515.3594 | 25.2353 | 5.7020 | 2.6462 | 1.7681 | 1.3805 | 1.209 | 1.119 | 1.0700 |
| | MADTM | 10% | 556.8970 | **30.4470** | **7.8988** | **3.9769** | **2.6700** | **2.0638** | **1.7382** | **1.5273** | **1.3888** |
| | | **20%** | **579.1679** | **28.7961** | **7.8069** | **4.0612** | **2.7952** | **2.1811** | **1.8388** | **1.6062** | **1.4558** |
| | | **25%** | **582.0973** | **31.3643** | **8.4908** | **4.3783** | **2.9698** | **2.2907** | **1.9347** | **1.6901** | **1.5214** |
| | MADWM | **10%** | **586.7648** | **26.0281** | **7.1988** | **3.8647** | **2.6895** | **2.1186** | **1.7936** | **1.5856** | **1.4362** |
| | | **20%** | **587.6499** | **25.9633** | **7.2035** | **3.8703** | **2.7042** | **2.119** | **1.7961** | **1.5798** | **1.4403** |
| | | **25%** | **609.9188** | **26.2598** | **7.2419** | **3.8887** | **2.7153** | **2.1253** | **1.8023** | **1.5866** | **1.4418** |

shows that *SDRL* has a significantly lower value than their targeted value for certain *CUSUM*-$\hat{\vartheta}$ chart and *SDRL* decreases for all charts as to the $\delta$ increases.

**5.1.2 Symmetric variance environment.** A symmetric variance distribution is used when the spread parameter has been disturbed. In such an environment, we examined the performance of the suggested estimators with their corresponding *CUSUM* charts in which each observation has a 99% probability that is derived from normal distribution N(0,1) and 1% probability taken from normal distribution $N(0,9)$. Tables 7 and 8 present the *ARL* and *SDRL* results of symmetric variance for sample sizes $n = 5$ *and* $n = 9$.

From Tables 7 and 8 results of *ARL* and *SDRL* show that *S* and *IQR* are better than *MADM* based on *CUSUM*-$\hat{\vartheta}$ charts of sample size $n = 5$ but less efficient for large sample size ($n = 9$). The larger values of *ARL* are highlighted. The *MDTM* (at 10%, 20%, and 25% trimming) and *MDWM* (at 10%, 20%, and 25% winsorizing) are reasonably good performances at the sample size $n = 9$ although they are more efficient than *S*, *IQR*, and *MADM*. The proposed estimator

**Table 6. *SDRL* values of robust estimators based on *CUSUM-$\hat{\vartheta}$* charts in uncontaminated environment N(0,1) when $ARL_O = 500$.**

| n | Estimator | α | δ | | | | | | | | |
|---|---|---|---|---|---|---|---|---|---|---|---|
| | | | **1.00** | **1.25** | **1.50** | **1.75** | **2.00** | **2.25** | **2.50** | **2.75** | **3.00** |
| 5 | S | | 494.0566 | 13.0486 | 3.6699 | 1.9940 | 1.3855 | 1.0914 | 0.9051 | 0.7808 | 0.6937 |
| | IQR | | 388.2293 | 47.0810 | 14.5926 | 7.1143 | 4.2397 | 3.0054 | 2.2632 | 1.8017 | 1.5398 |
| | MADM | | 389.7046 | 24.3560 | 8.0053 | 4.4279 | 3.0701 | 2.3707 | 1.9814 | 1.7176 | 1.5137 |
| | MDTM | 10% | 428.7824 | 26.6132 | 6.5782 | 2.9351 | 1.7470 | 1.2350 | 0.9338 | 0.7550 | 0.6275 |
| | | 20% | 519.3917 | 38.3522 | 9.2358 | 3.9808 | 2.2372 | 1.4845 | 1.0959 | 0.8588 | 0.7079 |
| | | 25% | 519.3917 | 38.3522 | 9.2358 | 3.9808 | 2.2372 | 1.4845 | 1.0959 | 0.8588 | 0.7079 |
| | MDWM | 10% | 425.0279 | 34.2181 | 8.4814 | 3.7604 | 2.1264 | 1.4122 | 1.0384 | 0.8227 | 0.6762 |
| | | 20% | 434.9548 | 37.5941 | 9.8179 | 4.2667 | 2.3962 | 1.5644 | 1.1419 | 0.8819 | 0.7239 |
| | | 25% | 433.3795 | 37.5156 | 9.7206 | 4.2613 | 2.3808 | 1.5672 | 1.1346 | 0.8792 | 0.7235 |
| | MADTM | 10% | 527.3098 | 28.8370 | 7.7651 | 3.7717 | 2.4392 | 1.7990 | 1.4388 | 1.2225 | 1.0772 |
| | | **20%** | **566.7940** | **52.2289** | **14.6974** | **6.7484** | **4.0767** | **2.8135** | **2.1156** | **1.7162** | **1.4718** |
| | | **25%** | **570.3901** | **52.4685** | **14.7666** | **6.7830** | **4.0857** | **2.8301** | **2.1212** | **1.7211** | **1.4734** |
| | MADWM | **10%** | **569.9133** | **39.2470** | **10.1379** | **4.7666** | **2.9083** | **2.0394** | **1.6066** | **1.3288** | **1.1423** |
| | | 20% | 230.2351 | 35.0984 | 12.2316 | 6.1730 | 3.8081 | 2.6235 | 1.9694 | 1.5730 | 1.3108 |
| | | 25% | 232.4507 | 35.8688 | 12.5086 | 6.2710 | 3.9011 | 2.6999 | 2.0251 | 1.6320 | 1.3574 |
| 9 | S | | 437.1928 | 6.71873 | 1.8769 | 1.0847 | 0.7800 | 0.6184 | 0.5079 | 0.4100 | 0.3318 |
| | IQR | | 461.1990 | 31.3513 | 7.8131 | 3.5583 | 2.0818 | 1.4791 | 1.1420 | 0.9223 | 0.7711 |
| | MADM | | 398.5663 | 15.5313 | 4.6162 | 2.4946 | 1.7119 | 1.3394 | 1.1141 | 0.9607 | 0.8553 |
| | MDTM | 10% | 546.4352 | 20.2956 | 3.8617 | 1.6548 | 0.9641 | 0.6510 | 0.4682 | 0.3481 | 0.2684 |
| | | 20% | 546.5548 | 21.6604 | 4.1389 | 1.7517 | 1.0067 | 0.6662 | 0.4808 | 0.3588 | 0.2726 |
| | | **25%** | **575.5918** | 21.4343 | 4.0157 | 1.7111 | 0.9956 | 0.6656 | 0.4866 | 0.3628 | 0.2787 |
| | MDWM | 10% | 555.9454 | 21.2728 | 3.9851 | 1.6943 | 0.9892 | 0.6567 | 0.4745 | 0.3526 | 0.2716 |
| | | 20% | 543.8822 | 21.1574 | 4.0517 | 1.7090 | 0.9898 | 0.6602 | 0.4750 | 0.3552 | 0.2713 |
| | | 25% | 514.3754 | 24.8902 | 5.1420 | 2.1041 | 1.1593 | 0.7304 | 0.5104 | 0.3686 | 0.2741 |
| | MADTM | 10% | 548.2321 | **29.0414** | **6.7057** | **2.9489** | **1.7905** | **1.2524** | **0.9603** | **0.7829** | **0.6540** |
| | | **20%** | **574.6650** | **27.0759** | **6.3863** | **2.8530** | **1.7869** | **1.2867** | **1.0062** | **0.8220** | **0.6913** |
| | | **25%** | **578.4091** | **29.9679** | **7.0411** | **3.1869** | **1.9635** | **1.3847** | **1.0922** | **0.8979** | **0.7590** |
| | MADWM | **10%** | **584.8985** | **24.1742** | **5.6144** | **2.5976** | **1.6273** | **1.1851** | **0.9306** | **0.7771** | **0.6633** |
| | | **20%** | **586.0281** | **24.0385** | **5.6364** | **2.5782** | **1.6363** | **1.1837** | **0.9308** | **0.7729** | **0.6667** |
| | | **25%** | **609.4564** | **24.5047** | **5.6427** | **2.5833** | **1.6415** | **1.1908** | **0.9391** | **0.7749** | **0.6641** |

*MADTM* (at 10%, 20%, and 25% trimming) for both sample sizes *n* = 5 *and n* = 9 has shown best overall performance than other estimators for all shifts of the dispersion process. The *MADWM* (at 10%, 20%, and 25% winsorizing) is very sensitive when the sample size is small *n* = 5 but as the sample size increases (*n* = 9) the *MADWM* (at 10%, 20% and 25% winsorizing) performs well as compared to *S, IQR,* and *MADM*. The shift *δ* > 1.25 the *IQR*, the *MADTM* (at 20% and 25% trimming) and *MADWM* (at 10% winsorizing) are good for small sample size *n* = 5 when the sample size is large *n* = 9 the *MADTM* (at 10%, 20%, and 25% trimming) and *MADWM* (at 10%, 20% and 25% winsorizing) performs best as compared to other estimators in the increasing shift of the dispersion process.

**5.1.3 Asymmetric variance environment.** In an asymmetric variance environment, each observation is taken from normal distribution *N*(0,1) and has a 1% probability of adding a multiple of $\chi_1^2$ Chi-Square with one degree of freedom to it with a multiplier equal to 4. Tables 9 and 10 show the results of *ARL* and *SDRL* respectively for sample sizes *n* = 5 and *n* = 9.

**Table 7. ARL values of robust estimators based on $CUSUM$-$\hat{\vartheta}$ charts under symmetric variance contaminated environment when $ARL_O$ = 500.**

| n | Estimator | α | δ | | | | | | | | |
|---|---|---|---|---|---|---|---|---|---|---|---|
| | | | **1.00** | **1.25** | **1.50** | **1.75** | **2.00** | **2.25** | **2.50** | **2.75** | **3.00** |
| 5 | S | | 541.1164 | 20.9660 | 6.1532 | 3.4980 | 2.4867 | 1.9811 | 1.6973 | 1.5093 | 1.3873 |
| | IQR | | 519.2442 | 64.6708 | 20.4155 | 9.9811 | 6.1617 | 4.3762 | 3.3878 | 2.7893 | 2.4178 |
| | MADM | | 296.0217 | 39.5086 | 13.5180 | 7.1621 | 4.7846 | 3.6113 | 2.9310 | 2.5016 | 2.2004 |
| | MDTM | 10% | 431.6535 | 36.6848 | 9.8500 | 4.6016 | 2.8768 | 2.1222 | 1.7208 | 1.4980 | 1.3640 |
| | | 20% | 494.2060 | 40.3788 | 10.6988 | 4.9603 | 3.0485 | 2.2237 | 1.7950 | 1.5406 | 1.3969 |
| | | 25% | 495.3582 | 40.5918 | 10.7731 | 4.9907 | 3.0630 | 2.2342 | 1.8023 | 1.5430 | 1.3984 |
| | MDWM | 10% | 438.6991 | 37.6261 | 10.1637 | 4.7352 | 2.9376 | 2.1533 | 1.7437 | 1.5123 | 1.3758 |
| | | 20% | 480.4918 | 39.9682 | 10.6739 | 4.9159 | 3.0283 | 2.2058 | 1.7843 | 1.5315 | 1.3907 |
| | | 25% | 482.6789 | 40.1526 | 10.7224 | 4.9343 | 3.0503 | 2.2163 | 1.7922 | 1.5380 | 1.3948 |
| | MADTM | **10%** | *602.3176* | *45.9237* | *12.6018* | *6.2465* | *4.0746* | *3.0598* | *2.5010* | *2.1408* | *1.8988* |
| | | **20%** | *607.4826* | *58.6833* | *16.9151* | *8.2047* | *5.1717* | *3.7791* | *3.0229* | *2.5444* | *2.2341* |
| | | **25%** | *616.8357* | *58.1955* | *16.6428* | *8.0843* | *5.1265* | *3.7586* | *3.0103* | *2.5387* | *2.2330* |
| | MADWM | **10%** | *571.9853* | *41.2259* | *11.8382* | *6.1631* | *4.1313* | *3.1582* | *2.6052* | *2.2385* | *1.9993* |
| | | 20% | 230.4432 | 35.6443 | 12.6663 | 6.6794 | 4.3502 | 3.1983 | 2.5568 | 2.1839 | 1.9123 |
| | | 25% | 232.5157 | 36.3084 | 12.9369 | 6.8188 | 4.4543 | 3.2653 | 2.6088 | 2.2283 | 1.9572 |
| 9 | S | | 507.3033 | 9.65183 | 3.82967 | 2.5015 | 1.9243 | 1.5893 | 1.3866 | 1.2438 | 1.1577 |
| | IQR | | 553.2200 | 45.4744 | 11.9076 | 5.4683 | 3.3685 | 2.4642 | 2.0000 | 1.7113 | 1.5225 |
| | MADM | | 572.0332 | 33.2390 | 8.9681 | 4.6224 | 3.1227 | 2.4111 | 2.0162 | 1.7559 | 1.5772 |
| | MDTM | 10% | 598.9702 | 21.4217 | 4.7491 | 2.4138 | 1.6932 | 1.3732 | 1.2097 | 1.1222 | 1.0744 |
| | | **20%** | **608.4783** | 23.0338 | 5.0091 | 2.4830 | 1.7244 | 1.3801 | 1.2154 | 1.1268 | 1.0766 |
| | | **25%** | **668.1703** | 24.8277 | 5.2273 | 2.5528 | 1.7510 | 1.3978 | 1.2263 | 1.1318 | 1.0813 |
| | MDWM | 10% | 559.9072 | 21.7714 | 4.8333 | 2.4234 | 1.6980 | 1.3673 | 1.2070 | 1.1194 | 1.0723 |
| | | **20%** | **625.6945** | 22.9273 | 4.9722 | 2.4723 | 1.7220 | 1.3831 | 1.2159 | 1.1256 | 1.0760 |
| | | **25%** | **630.9913** | 23.4878 | 5.0437 | 2.4890 | 1.7298 | 1.3838 | 1.2177 | 1.1253 | 1.0756 |
| | MADTM | **10%** | **614.0313** | *26.9333* | *7.3037* | *3.8926* | *2.7103* | *2.1263* | *1.7977* | *1.5886* | *1.4388* |
| | | **20%** | **672.2903** | *30.0616* | *7.9963* | *4.1589* | *2.8553* | *2.2206* | *1.8723* | *1.6356* | *1.4778* |
| | | **25%** | **688.6003** | *34.4576* | *8.9538* | *4.5283* | *3.0447* | *2.3396* | *1.9653* | *1.7141* | *1.5373* |
| | MADWM | **10%** | **626.6833** | *27.6806* | *7.4183* | *3.9175* | *2.7068* | *2.1238* | *1.7948* | *1.5824* | *1.4324* |
| | | **20%** | **628.9926** | *27.5048* | *7.4424* | *3.9305* | *2.7186* | *2.1213* | *1.7955* | *1.5793* | *1.4351* |
| | | **25%** | **684.5610** | *28.4764* | *7.5354* | *3.9705* | *2.7452* | *2.1405* | *1.8097* | *1.5908* | *1.4433* |

The above Table 9 of *ARL* clearly illustrates that for a small sample size *n* = 5 the *S* and *MADM* are better than *MADWM* (at 20% and 25% winsorizing) but less efficient than the other estimators. When the sample size is small i.e *n* = 5, *IQR* performance is good based on *CUSUM*-$\hat{\vartheta}$ charts as compared to *S*, *MADM*, *MDTM* (at 10% trimming) and *MADWM* (at 20% and 25% winsorizing). The larger values of *ARL* are highlighted. For a small sample size *n* = 5 *MDTM* (at 20% and 25% trimming) is better than *S*, *IQR MADM*. For a large sample size *n* = 9 is better than *IQR* and *MADM*. The performance of *MDTM* (at 10%, 20%, and 25% trimming), *MDWM* (at 10%, 20%, and 25% winsorizing), and *MADWM* (at 10%, 20% and 25% winsorizing) is best for large sample size *n* = 9 and more efficient as compared to *S*, *IQR*, and *MADM*. The *MADTM* (at 10%, 20%, and 25% trimming) shows superior performance to other estimators in increasing all shifts of the dispersion process for both sample sizes *n* = 5 *and n* = 9. When *δ* > 1.25 *IQR*, *MADM* and *MADTM* (at 20%, and 25% trimming) outperform all other estimators for both sample sizes of *n* = 5 *and n* = 9.

**Table 8. SDRL values of robust estimators based on $CUSUM$-$\hat{\vartheta}$ charts under symmetric variance contaminated environment when $ARL_O$ = 500.**

| n | Estimator | α | δ | | | | | | | | |
|---|---|---|---|---|---|---|---|---|---|---|---|
| | | | **1.00** | **1.25** | **1.50** | **1.75** | **2.00** | **2.25** | **2.50** | **2.75** | **3.00** |
| 5 | S | | 535.7153 | 18.9487 | 4.6057 | 2.2605 | 1.4665 | 1.0831 | 0.8638 | 0.7215 | 0.6212 |
| | IQR | | 518.3827 | 64.0671 | 20.1737 | 9.5824 | 5.5799 | 3.8241 | 2.8165 | 2.1987 | 1.8276 |
| | MADM | | 295.9177 | 38.7367 | 12.7311 | 6.2207 | 3.9125 | 2.7926 | 2.1721 | 1.7735 | 1.4891 |
| | MDTM | 10% | 429.1784 | 36.3046 | 9.3753 | 4.0864 | 2.3043 | 1.5297 | 1.1080 | 0.8544 | 0.7021 |
| | | 20% | 492.3805 | 40.0812 | 10.2735 | 4.4494 | 2.4920 | 1.6239 | 1.1880 | 0.9064 | 0.7425 |
| | | 25% | 493.4163 | 40.2437 | 10.3649 | 4.4860 | 2.5084 | 1.6391 | 1.2008 | 0.9121 | 0.7468 |
| | MDWM | 10% | 436.7832 | 37.3552 | 9.7033 | 4.2577 | 2.3892 | 1.5587 | 1.1337 | 0.8752 | 0.7202 |
| | | 20% | 478.4572 | 39.6400 | 10.2827 | 4.4221 | 2.4726 | 1.6149 | 1.1792 | 0.8980 | 0.7371 |
| | | 25% | 481.2845 | 39.7313 | 10.3665 | 4.4560 | 2.4969 | 1.6273 | 1.1875 | 0.9075 | 0.7432 |
| | MADTM | **10%** | *601.7392* | *44.8926* | *11.2589* | *5.1089* | *3.0392* | *2.0958* | *1.6138* | *1.3147* | *1.1023* |
| | | **20%** | *605.3687* | *57.7511* | *16.1273* | *7.4251* | *4.3450* | *3.0080* | *2.2447* | *1.7950* | *1.5246* |
| | | **25%** | *615.7205* | *57.7205* | *15.8152* | *7.2441* | *4.2703* | *2.9618* | *2.2070* | *1.7659* | *1.5054* |
| | MADWM | **10%** | *569.9133* | *39.2470* | *10.1379* | *4.7666* | *2.9083* | *2.0394* | *1.6066* | *1.3288* | *1.1423* |
| | | 20% | 230.2351 | 35.0984 | 12.2316 | 6.1730 | 3.8081 | 2.6235 | 1.9694 | 1.5730 | 1.3108 |
| | | 25% | 232.4507 | 35.8688 | 12.5086 | 6.2800 | 3.9011 | 2.6999 | 2.0251 | 1.6320 | 1.3574 |
| 9 | S | | 503.2276 | 6.4361 | 1.8641 | 1.0849 | 0.8057 | 0.6454 | 0.5459 | 0.4541 | 0.3746 |
| | IQR | | 549.5134 | 44.5422 | 11.218 | 4.8617 | 2.7268 | 1.8197 | 1.3483 | 1.0565 | 0.8571 |
| | MADM | | 570.2915 | 31.6422 | 7.6065 | 3.4449 | 2.0857 | 1.4843 | 1.1696 | 0.9556 | 0.8099 |
| | MDTM | 10% | 591.6539 | 20.6956 | 3.85450 | 1.6434 | 0.9637 | 0.6557 | 0.4749 | 0.3546 | 0.2734 |
| | | 20% | 601.8185 | 22.4379 | 4.1561 | 1.7524 | 1.0108 | 0.6726 | 0.4851 | 0.3643 | 0.2791 |
| | | 25% | 664.0351 | 24.1573 | 4.3708 | 1.8148 | 1.0357 | 0.6917 | 0.4993 | 0.3720 | 0.2869 |
| | MDWM | 10% | 555.9454 | 21.2728 | 3.9851 | 1.6943 | 0.9892 | 0.6567 | 0.4745 | 0.3526 | 0.2716 |
| | | 20% | 620.2006 | 22.2399 | 4.1162 | 1.7341 | 1.0036 | 0.6721 | 0.4832 | 0.3619 | 0.2776 |
| | | 25% | 626.5939 | 22.7383 | 4.2188 | 1.7641 | 1.0167 | 0.6754 | 0.4889 | 0.3620 | 0.2770 |
| | MADTM | 10% | 611.0962 | *25.0070* | *5.7165* | *2.6254* | *1.6279* | *1.1918* | *0.9392* | *0.7853* | *0.6689* |
| | | **20%** | 667.0722 | *28.3341* | *6.4915* | *2.8931* | *1.8011* | *1.2963* | *1.0156* | *0.8347* | *0.7023* |
| | | **25%** | 688.2372 | *33.0933* | *7.4693* | *3.3044* | *2.0190* | *1.4297* | *1.1171* | *0.9208* | *0.7729* |
| | MADWM | 10% | *623.3632* | *25.7678* | *5.9212* | *2.6796* | *1.6663* | *1.2039* | *0.9393* | *0.7818* | *0.6647* |
| | | 20% | *623.6745* | *25.8289* | *5.9317* | *2.6749* | *1.6689* | *1.2016* | *0.9429* | *0.7790* | *0.6656* |
| | | 25% | *676.4403* | *26.6824* | *5.9667* | *2.6881* | *1.6817* | *1.2158* | *0.9517* | *0.7824* | *0.6677* |

**5.1.4 Non-normal environment.** The samples prepared in this way are transformed without loss of generality. One way to get the resulting sample with zero mean and one variance. For this reason, the mean is subtracted from each sample taken from the non-normal environment and then divided by the non-normal environment of the standard deviation to determine the correct result and comparable performance.

Tables 11 and 12 present the *ARL* values of different estimators to predict an increase in dispersion process at different magnitudes for in-control $ARL_O$ = 500 and sample size $n$ = 5 when underlying process distribution are Gamma and Logistic. The following are some important outcomes of *ARL* and *SDRL* values of Gamma distribution G(2,1).

The *MADM* and the *MDTM* (at 10%, 20%, and 25% trimming) show good performance as compared to *S* and *IQR*. The *MDWM* (at 10%, 20%, and 25% winsorizing) performance is better than *S*, *IQR*, *MADM*, *MDTM*(at 10%, 20%, and 25% trimming) and *MADTM* (10% trimming). The performance of the proposed estimator *MADTM* (10%, 20%, and 25% trimming)

**Table 9. ARL values of robust estimators based on $CUSUM$-$\hat{\vartheta}$ charts under asymmetric variance contaminated environment when $ARL_O = 500$.**

| n | Estimator | α | δ | | | | | | | | |
|---|-----------|---|------|------|------|------|------|------|------|------|------|
| | | | 1.00 | 1.25 | 1.50 | 1.75 | 2.00 | 2.25 | 2.50 | 2.75 | 3.00 |
| 5 | S | | 307.7253 | 23.1488 | 6.5588 | 3.4046 | 2.3062 | 1.8043 | 1.5318 | 1.3768 | 1.2723 |
| | IQR | | 453.8029 | 57.7373 | 18.3389 | 9.0975 | 5.7141 | 4.0874 | 3.2142 | 2.6625 | 2.3189 |
| | MADM | | 296.0217 | 39.5086 | 13.5180 | 7.1621 | 4.7846 | 3.6113 | 2.9310 | 2.5016 | 2.2004 |
| | MDTM | 10% | 431.7089 | 36.6988 | 9.8639 | 4.6083 | 2.8787 | 2.1246 | 1.7215 | 1.4984 | 1.3643 |
| | | 20% | 494.6059 | 40.4333 | 10.7301 | 4.9706 | 3.0531 | 2.2273 | 1.7977 | 1.5414 | 1.3974 |
| | | 25% | 495.3582 | 40.5918 | 10.7731 | 4.9907 | 3.0630 | 2.2342 | 1.8023 | 1.5430 | 1.3984 |
| | MDWM | 10% | 476.1050 | 39.4377 | 10.5170 | 4.8451 | 2.9872 | 2.1843 | 1.7630 | 1.5210 | 1.3833 |
| | | 20% | 484.9236 | 40.1528 | 10.7090 | 4.9283 | 3.0341 | 2.2085 | 1.7861 | 1.5325 | 1.3916 |
| | | 25% | 486.8718 | 40.3788 | 10.7633 | 4.9515 | 3.0580 | 2.2208 | 1.7944 | 1.5402 | 1.3953 |
| | MADTM | **10%** | **602.3176** | 45.9237 | 12.6018 | 6.2465 | 4.0746 | 3.0598 | 2.5010 | 2.1408 | 1.8988 |
| | | **20%** | **607.7727** | *57.2463* | *16.4913* | *8.0116* | *5.0992* | *3.7407* | *2.9988* | *2.5348* | *2.2292* |
| | | **25%** | **616.8357** | *58.1955* | *16.6428* | *8.0843* | *5.1265* | *3.7586* | *3.0103* | *2.5387* | *2.2330* |
| | MADWM | **10%** | **586.7783** | 41.8368 | 11.9664 | 6.1907 | 4.1499 | 3.1706 | 2.6128 | 2.2431 | 2.0037 |
| | | 20% | 230.4432 | 35.6443 | 12.6663 | 6.6794 | 4.3502 | 3.1983 | 2.5568 | 2.1839 | 1.9123 |
| | | 25% | 232.5157 | 36.3084 | 12.9369 | 6.8188 | 4.4543 | 3.2653 | 2.6088 | 2.2283 | 1.9572 |
| 9 | S | | 527.6334 | 9.7398 | 3.86467 | 2.5198 | 1.9355 | 1.5988 | 1.3927 | 1.2499 | 1.1610 |
| | IQR | | 419.8958 | 34.5637 | 9.5670 | 4.6599 | 3.0155 | 2.2820 | 1.8978 | 1.6461 | 1.4868 |
| | MADM | | 449.6042 | 27.6108 | 8.0005 | 4.3434 | 3.0105 | 2.3663 | 1.9896 | 1.7428 | 1.5707 |
| | MDTM | 10% | 557.3967 | 21.4825 | 4.7982 | 2.4106 | 1.6875 | 1.3633 | 1.2033 | 1.1171 | 1.0716 |
| | | 20% | 597.4698 | 22.4510 | 4.9403 | 2.4634 | 1.7175 | 1.3783 | 1.2149 | 1.1263 | 1.0764 |
| | | **25%** | **645.3178** | 23.3315 | 5.0295 | 2.5062 | 1.7368 | 1.3931 | 1.2253 | 1.1312 | 1.0811 |
| | MDWM | 10% | 550.3908 | 21.1610 | 4.7561 | 2.4000 | 1.6908 | 1.3656 | 1.2066 | 1.1193 | 1.0723 |
| | | 20% | 559.7048 | 21.7599 | 4.8477 | 2.4273 | 1.7003 | 1.3687 | 1.2086 | 1.1209 | 1.0729 |
| | | 25% | 621.2065 | 22.8415 | 4.9675 | 2.4701 | 1.7213 | 1.3818 | 1.2171 | 1.1250 | 1.0756 |
| | MADTM | **10%** | **605.2809** | *26.2787* | *7.2293* | *3.8783* | *2.7064* | *2.1286* | *1.8028* | *1.5923* | *1.4428* |
| | | **20%** | **607.6565** | *28.5506* | *7.7488* | *4.0724* | *2.8116* | *2.2009* | *1.8603* | *1.6249* | *1.4716* |
| | | **25%** | **652.1019** | *31.9874* | *8.5739* | *4.4378* | *3.0164* | *2.3371* | *1.9687* | *1.7199* | *1.5453* |
| | MADWM | **10%** | **595.0235** | 26.6240 | 7.2638 | 3.8796 | 2.6878 | 2.1148 | 1.7893 | 1.5808 | 1.4315 |
| | | **20%** | **600.0943** | 27.0467 | 7.3790 | 3.9033 | 2.7011 | 2.1121 | 1.7860 | 1.5731 | 1.4302 |
| | | **25%** | **626.0596** | 27.4518 | 7.4087 | 3.9174 | 2.7113 | 2.1166 | 1.7898 | 1.5778 | 1.4345 |

is best as compared to all other estimators in the increasing shifts of the dispersion process. The larger values of *ARL* are highlighted. The *MADWM* (at 10%, 20%, and 25% winsorizing) perform work well as compared to *S*, *IQR*, *MADM*, *MDTM*(at 10%, 20%, and 25% trimming), *MDWM* (at 10%, 20% and 25% winsorizing). For $\delta > 1.25$ the *IQR*, *MADM*, *MADTM*(at 20% and 25% trimming) and *MADWM* (at 20% and 25% winsorizing) performs work well as compared to other estimators.

Tables 13 and 14 present the *ARL* and *SDRL* values based on $CUSUM$-$\hat{\vartheta}$ charts of Logistic distribution.

For the logistic distribution, the *MADM* performs work well as compared to *S*, *IQR*, and *MDTM* (at 10%, 20% trimming) and *MADWM* (at 10%, 20% and 25% winsorizing). The *MDWM* (at 10%, 20%, and 25% winsorizing) performance is better than *S*, *IQR*, *MADM*, *MDTM*(at 10% and 20%, 25% trimming), *MADTM* (10% trimming) and *MADWM* (at 10%, 20% and 25% winsorizing). The proposed estimator *MADTM* (at 10%, 20%, and 25%

**Table 10. SDRL values of robust estimators based on $CUSUM$-$\hat{\vartheta}$ charts under asymmetric variance contaminated environment when $ARL_O = 500$.**

| $n$ | Estimator | $\alpha$ | $\delta$ | | | | | | | | |
|---|---|---|---|---|---|---|---|---|---|---|---|
| | | | 1.00 | 1.25 | 1.50 | 1.75 | 2.00 | 2.25 | 2.50 | 2.75 | 3.00 |
| 5 | S | | 306.5967 | 22.7350 | 5.8635 | 2.7357 | 1.6312 | 1.1331 | 0.8513 | 0.6834 | 0.5658 |
| | IQR | | 453.3361 | 57.3511 | 17.9345 | 8.6593 | 5.0887 | 3.4984 | 2.6178 | 2.0359 | 1.7023 |
| | MADM | | 295.9177 | 38.7367 | 12.7311 | 6.2207 | 3.9125 | 2.7926 | 2.1721 | 1.7735 | 1.4891 |
| | MDTM | 10% | 429.1916 | 36.3264 | 9.3919 | 4.0953 | 2.3065 | 1.5327 | 1.1088 | 0.8552 | 0.7031 |
| | | 20% | 492.6785 | 40.1681 | 10.3025 | 4.4634 | 2.4992 | 1.6294 | 1.1930 | 0.9085 | 0.7439 |
| | | 25% | 493.4163 | 40.2437 | 10.3649 | 4.4860 | 2.5084 | 1.6391 | 1.2008 | 0.9121 | 0.7468 |
| | MDWM | 10% | 475.3080 | 39.1457 | 10.0787 | 4.3498 | 2.4363 | 1.5964 | 1.1535 | 0.8861 | 0.7293 |
| | | 20% | 483.2781 | 39.7779 | 10.3434 | 4.4338 | 2.4769 | 1.6168 | 1.1806 | 0.8993 | 0.7378 |
| | | 25% | 485.3560 | 39.9031 | 10.4108 | 4.4706 | 2.5045 | 1.6314 | 1.1923 | 0.9098 | 0.7438 |
| | MADTM | **10%** | **601.7392** | 44.8926 | 11.2589 | 5.1089 | 3.0392 | 2.0982 | 1.6138 | 1.3147 | 1.1023 |
| | | **20%** | **607.5638** | *56.0603* | *15.6448* | *7.1587* | *4.2299* | *2.9353* | *2.1904* | *1.7591* | *1.5002* |
| | | **25%** | **615.7205** | *57.0553* | *15.8152* | *7.2441* | *4.2703* | *2.9618* | *2.2070* | *1.7659* | *1.5054* |
| | MADWM | **10%** | **585.7029** | 39.8143 | 10.2567 | 4.7949 | 2.9193 | 2.0522 | 1.6119 | 1.3311 | 1.1462 |
| | | 20% | 230.2351 | 35.0984 | 12.2316 | 6.1730 | 3.8081 | 2.6235 | 1.9694 | 1.5730 | 1.3108 |
| | | 25% | 232.4507 | 35.8688 | 12.5086 | 6.2800 | 3.9011 | 2.6999 | 2.0251 | 1.6320 | 1.3574 |
| 9 | S | | 523.1535 | 6.4826 | 1.8777 | 1.0887 | 0.8108 | 0.6488 | 0.5491 | 0.4586 | 0.3783 |
| | IQR | | 419.5338 | 33.7016 | 8.7040 | 3.9172 | 2.2642 | 1.5642 | 1.1955 | 0.9492 | 0.7906 |
| | MADM | | 447.8255 | 25.8888 | 6.5124 | 3.0714 | 1.9345 | 1.4127 | 1.1129 | 0.9233 | 0.7868 |
| | MDTM | 10% | 555.2687 | 20.9262 | 3.9505 | 1.6760 | 0.9756 | 0.6537 | 0.4699 | 0.3491 | 0.2698 |
| | | 20% | 594.3667 | 21.7819 | 4.0582 | 1.7191 | 1.0004 | 0.6659 | 0.4835 | 0.3625 | 0.2782 |
| | | **25%** | **642.0142** | 22.4905 | 4.1197 | 1.7414 | 1.0120 | 0.6786 | 0.4952 | 0.3692 | 0.2861 |
| | MDWM | 10% | 548.5753 | 20.5468 | 3.9023 | 1.6613 | 0.9739 | 0.6538 | 0.4729 | 0.3520 | 0.2716 |
| | | 20% | 557.4638 | 21.0981 | 4.0120 | 1.6998 | 0.9875 | 0.6594 | 0.4762 | 0.3563 | 0.2731 |
| | | 25% | 617.0015 | 22.0913 | 4.1151 | 1.7338 | 1.0027 | 0.6705 | 0.4853 | 0.3611 | 0.2770 |
| | MADTM | **10%** | **603.5732** | *24.2827* | *5.6061* | *2.5896* | *1.6144* | *1.1876* | *0.9350* | *0.7838* | *0.6684* |
| | | **20%** | **606.7028** | *26.7913* | *6.2213* | *2.8001* | *1.7534* | *1.2732* | *1.0052* | *0.8251* | *0.6966* |
| | | **25%** | **651.4412** | *30.3927* | *6.9873* | *3.1792* | *1.9566* | *1.3968* | *1.1027* | *0.9111* | *0.7695* |
| | MADWM | **10%** | **592.3477** | 24.7761 | 5.7126 | 2.6363 | 1.6415 | 1.1899 | 0.9309 | 0.7791 | 0.6617 |
| | | **20%** | **596.4870** | 25.3300 | 5.8815 | 2.6552 | 1.6635 | 1.1989 | 0.9359 | 0.7751 | 0.6635 |
| | | **25%** | **620.2770** | 25.7279 | 5.9156 | 2.6616 | 1.6674 | 1.2016 | 0.9423 | 0.7765 | 0.6618 |

trimming) performance is excellent as compared to all other estimators in increasing shifts of the dispersion process. The *IQR*, *MADM MADTM* (at 10%, 20%, and 25% trimming) and *MADWM* (at 20% winsorizing) perform work well for $\delta > 1.25$.

## 6 Conclusion

In this paper, several estimators of dispersion parameters are considered for use in the development of Phase II control limits. These include some widely used estimators as well as robust estimators that are uncommon in the literature of control charts. The robust dispersion parameter was monitored using the $CUSUM$-$\hat{\vartheta}$ control chart structure for these estimators. In different environments, the results of these robust estimators are evaluated. The uncontaminated environment, different contaminated environments symmetric variance, asymmetric variance disturbances and non-normal environments. All charts perform well under the

**Table 11. ARL values of robust estimators based on $CUSUM\text{-}\hat{\vartheta}$ charts under G(2,1) environment when $ARL_O = 500$.**

| $n$ | Estimator | $\alpha$ | $\delta$ | | | | | | | | |
|---|---|---|---|---|---|---|---|---|---|---|---|
| | | | 1.00 | 1.25 | 1.50 | 1.75 | 2.00 | 2.25 | 2.50 | 2.75 | 3.00 |
| 5 | S | | 516.1774 | 87.7922 | 26.9402 | 12.0510 | 6.7903 | 4.5060 | 3.3868 | 2.6972 | 2.2823 |
| | IQR | | 519.7193 | 95.4686 | 31.6550 | 15.1649 | 9.0925 | 6.2743 | 4.7368 | 3.8324 | 3.2541 |
| | MADM | | 551.0617 | 99.5754 | 33.1752 | 15.8869 | 9.4089 | 6.3986 | 4.8020 | 3.8223 | 3.2166 |
| | MDTM | 10% | 571.2206 | 74.5593 | 20.4720 | 9.0590 | 5.3742 | 3.8053 | 2.9553 | 2.4461 | 2.1194 |
| | | 20% | 580.1879 | 63.7846 | 17.0788 | 7.7118 | 4.7128 | 3.4141 | 2.6773 | 2.2600 | 1.9762 |
| | | 25% | **587.6944** | 65.0594 | 17.3653 | 7.7868 | 4.7411 | 3.4332 | 2.6879 | 2.2685 | 1.9813 |
| | MDWM | 10% | 579.5244 | 78.6777 | 21.9607 | 9.6578 | 5.6018 | 3.9010 | 2.9853 | 2.4495 | 2.1118 |
| | | 20% | **586.8547** | 75.4348 | 20.5690 | 9.0725 | 5.3108 | 3.7488 | 2.8771 | 2.3921 | 2.0638 |
| | | 25% | **588.5643** | 75.8677 | 20.8288 | 9.1556 | 5.3508 | 3.7619 | 2.8787 | 2.3881 | 2.0610 |
| | MADTM | 10% | 573.0771 | 78.7383 | 23.0849 | 10.6628 | 6.3712 | 4.4498 | 3.4348 | 2.8016 | 2.4096 |
| | | 20% | **587.2402** | **93.0562** | **29.1989** | **13.7920** | **8.1869** | **5.6571** | **4.3117** | **3.4997** | **2.9543** |
| | | 25% | **593.0874** | **94.5462** | **29.7139** | **14.0429** | **8.3147** | **5.7403** | **4.3576** | **3.5313** | **2.9752** |
| | MADWM | **10%** | **581.1239** | 82.4388 | 24.1996 | 11.0970 | 6.5997 | 4.6112 | 3.5456 | 2.8832 | 2.4756 |
| | | **20%** | **587.7783** | **89.4464** | **27.0191** | **12.3975** | **7.2631** | **5.0158** | **3.8119** | **3.1061** | **2.6359** |
| | | **25%** | **589.1387** | **91.6438** | **28.0711** | **12.9894** | **7.5953** | **5.2216** | **3.9447** | **3.2239** | **2.7243** |

uncontaminated environment, but the $CUSUM\text{-}\hat{\vartheta}$ control chart based on the MADTM (at 20% and 25% trimming) and MADWM (at 10%, 20% and 25% winsorizing) outperform all estimators under normality for large sample size $n = 9$. The performance of suggested estimators MDTM (at 10%, 20%, and 25% trimming) and MADTM (10%, 20%, and 25% trimming) are good for both sample sizes $n = 5$ and $n = 9$ in symmetric variance and asymmetric variance environment. When the environment is non-normal the estimators MDTM (at 25%

**Table 12. SDRL values of robust estimators based on $CUSUM\text{-}\hat{\vartheta}$ charts under G(2,1) environment when $ARL_O = 500$.**

| $n$ | Estimator | $\alpha$ | $\delta$ | | | | | | | | |
|---|---|---|---|---|---|---|---|---|---|---|---|
| | | | 1.00 | 1.25 | 1.50 | 1.75 | 2.00 | 2.25 | 2.50 | 2.75 | 3.00 |
| 5 | S | | 511.809 | 86.8856 | 26.0920 | 11.173 | 6.0533 | 3.7481 | 2.6411 | 1.9720 | 1.5808 |
| | IQR | | 509.585 | 95.4334 | 31.0556 | 14.280 | 8.3378 | 5.5645 | 4.0486 | 3.1654 | 2.5789 |
| | MADM | | 539.749 | 98.7306 | 32.8537 | 15.063 | 8.7228 | 5.7345 | 4.1601 | 3.1728 | 2.5968 |
| | MDTM | 10% | 571.112 | 73.9486 | 19.4088 | 7.8845 | 4.3174 | 2.8321 | 2.0258 | 1.5935 | 1.3145 |
| | | 20% | 578.912 | 62.1314 | 15.8577 | 6.5546 | 3.6433 | 2.4365 | 1.7773 | 1.4096 | 1.1787 |
| | | 25% | **585.3441** | 63.4501 | 16.1788 | 6.6366 | 3.6731 | 2.4670 | 1.7917 | 1.4205 | 1.1867 |
| | MDWM | 10% | 575.5705 | 77.1398 | 20.9713 | 8.6739 | 4.7137 | 3.0454 | 2.1559 | 1.6730 | 1.3625 |
| | | 20% | **579.903** | 74.2116 | 19.5463 | 8.0140 | 4.3659 | 2.8587 | 2.0271 | 1.5979 | 1.2975 |
| | | 25% | **582.631** | 74.2347 | 19.8326 | 8.1703 | 4.4676 | 2.9082 | 2.0567 | 1.6080 | 1.3085 |
| | MADTM | 10% | 562.9305 | 77.7594 | 22.4967 | 9.8338 | 5.5533 | 3.7026 | 2.7066 | 2.0666 | 1.6970 |
| | | 20% | **585.2164** | **93.6346** | **28.6986** | **12.8812** | **7.4442** | **4.9111** | **3.6484** | **2.8590** | **2.3028** |
| | | 25% | **592.3854** | **95.1506** | **29.3106** | **13.2082** | **7.6056** | **5.0381** | **3.7046** | **2.8996** | **2.3347** |
| | MADWM | **10%** | **571.6772** | 81.3094 | 23.3239 | 10.1881 | 5.7976 | 3.8652 | 2.7933 | 2.1513 | 1.7556 |
| | | **20%** | **576.897** | **89.1426** | **26.1374** | **11.6168** | **6.4842** | **4.2835** | **3.1097** | **2.3983** | **1.9277** |
| | | **25%** | **580.313** | **91.7491** | **27.1399** | **12.1638** | **6.7889** | **4.4816** | **3.2412** | **2.5033** | **2.0197** |

**Table 13. ARL values of robust estimators based on $CUSUM\text{-}\hat{\vartheta}$ charts under Logistic(2,1) environment when $ARL_O = 500$.**

| n | Estimator | α | δ | | | | | | | | |
|---|---|---|---|---|---|---|---|---|---|---|---|
| | | | 1.00 | 1.25 | 1.50 | 1.75 | 2.00 | 2.25 | 2.50 | 2.75 | 3.00 |
| 5 | S | | 523.0217 | 56.8409 | 14.3673 | 6.4413 | 3.8751 | 2.7684 | 2.2230 | 1.8784 | 1.6606 |
| | IQR | | 506.7088 | 73.4495 | 23.0666 | 11.0455 | 6.7669 | 4.7173 | 3.6456 | 3.0028 | 2.5845 |
| | MADM | | 581.3526 | 85.0645 | 26.5731 | 12.5860 | 7.4547 | 5.1724 | 3.9550 | 3.2470 | 2.7853 |
| | MDTM | 10% | 575.2277 | 52.6825 | 13.0355 | 5.9548 | 3.6682 | 2.6854 | 2.1666 | 1.8463 | 1.6428 |
| | | 20% | 580.2955 | 48.6720 | 12.1202 | 5.6585 | 3.5456 | 2.6352 | 2.1443 | 1.8373 | 1.6348 |
| | | **25%** | **587.9437** | 49.5341 | 12.3043 | 5.6948 | 3.5656 | 2.6448 | 2.1475 | 1.8396 | 1.6358 |
| | MDWM | **10%** | **586.5277** | 47.8238 | 11.8699 | 5.5768 | 3.5403 | 2.6331 | 2.1470 | 1.8418 | 1.6412 |
| | | **20%** | **590.3151** | 53.1881 | 13.0838 | 5.9967 | 3.6777 | 2.6876 | 2.1633 | 1.8445 | 1.6382 |
| | | **25%** | **600.5060** | 54.5199 | 13.4194 | 6.1022 | 3.7228 | 2.7090 | 2.1753 | 1.8524 | 1.6416 |
| | MADTM | **10%** | **586.0886** | *72.2994* | *20.4485* | *9.2704* | *5.5453* | *3.8196* | *2.9818* | *2.4846* | *2.1478* |
| | | **20%** | **599.9482** | *73.2398* | *21.1790* | *10.1039* | *6.1809* | *4.3998* | *3.4592* | *2.8516* | *2.4882* |
| | | **25%** | **605.7264** | *74.0268* | *21.4033* | *10.1720* | *6.2105* | *4.4123* | *3.4701* | *2.8589* | *2.4900* |
| | MADWM | 10% | 475.9947 | 58.4975 | 17.2604 | 8.1959 | 5.0744 | 3.6221 | 2.8807 | 2.4303 | 2.1208 |
| | | **20%** | 478.8376 | *69.9533* | *21.5248* | *10.1889* | *6.1155* | *4.2092* | *3.2197* | *2.6623* | *2.2925* |
| | | 25% | 501.1044 | 67.7446 | 20.2891 | 9.6414 | 5.8501 | 4.0971 | 3.1796 | 2.6513 | 2.2963 |

trimming), *MDWM* (at 10%, 20%, and 25% winsorizing), *MADTM* (10%, 20%, and 25% trimming) and *MADWM* (at 10%, 20% and 25% winsorizing) perform best under Gamma distribution. For Logistic distribution, the *MDTM* (at 25% trimming), *MDWM* (at 10%, 20%, and 25% winsorizing), and *MADTM* (10%, 20%, and 25% trimming) perform best than other estimators. In general, robust estimators *MADTM* (at 10%, 20%, and 25% trimming) and *MADWM* (10%, 20%, and 25% winsorizing) based on *CUSUM* charts perform superior in all

**Table 14. SDRL values of robust estimators based on $CUSUM\text{-}\hat{\vartheta}$ charts under Logistic(2,1) environment when $ARL_O = 500$.**

| n | Estimator | α | δ | | | | | | | | |
|---|---|---|---|---|---|---|---|---|---|---|---|
| | | | 1.00 | 1.25 | 1.50 | 1.75 | 2.00 | 2.25 | 2.50 | 2.75 | 3.00 |
| 5 | S | | 516.6046 | 55.4951 | 13.5514 | 5.5139 | 3.0032 | 1.9604 | 1.4312 | 1.1275 | 0.9344 |
| | IQR | | 503.9076 | 72.3001 | 22.1386 | 10.1618 | 6.0629 | 3.9786 | 2.9526 | 2.3458 | 1.9398 |
| | MADM | | 574.4848 | 84.8486 | 25.8869 | 11.6669 | 6.7494 | 4.4342 | 3.2439 | 2.5496 | 2.0733 |
| | MDTM | 10% | 567.9996 | 51.7144 | 11.8739 | 4.8885 | 2.7705 | 1.8461 | 1.3742 | 1.0763 | 0.9013 |
| | | 20% | 567.5776 | 47.6882 | 10.9060 | 4.5542 | 2.5925 | 1.7681 | 1.3273 | 1.0574 | 0.8826 |
| | | **25%** | **577.3358** | 48.4572 | 11.0964 | 4.6036 | 2.6147 | 1.7845 | 1.3333 | 1.0642 | 0.8857 |
| | MDWM | **10%** | **568.8170** | 46.4350 | 10.5778 | 4.4067 | 2.5312 | 1.7290 | 1.3104 | 1.0447 | 0.8728 |
| | | **20%** | **586.1505** | 52.1003 | 11.8829 | 4.9763 | 2.8095 | 1.8607 | 1.3764 | 1.0889 | 0.9028 |
| | | **25%** | **596.2466** | 53.4280 | 12.2994 | 5.1193 | 2.8769 | 1.9000 | 1.4040 | 1.1064 | 0.9132 |
| | MADTM | **10%** | **585.8793** | *71.0937* | *19.4379* | *8.2884* | *4.7675* | *3.0527* | *2.2491* | *1.7929* | *1.4579* |
| | | **20%** | **596.3077** | *72.0931* | *19.9656* | *8.9936* | *5.2861* | *3.5074* | *2.6674* | *2.0761* | *1.7582* |
| | | **25%** | **601.5520** | *72.5998* | *20.2237* | *9.1046* | *5.3228* | *3.5244* | *2.6780* | *2.0885* | *1.7608* |
| | MADWM | 10% | 469.6338 | 57.3827 | 16.0295 | 7.2135 | 4.1663 | 2.7553 | 2.0957 | 1.6797 | 1.3753 |
| | | **20%** | 475.7662 | *69.1187* | *20.4757* | *9.47046* | *5.4989* | *3.5245* | *2.5616* | *2.0285* | *1.6687* |
| | | 25% | 499.9614 | 66.9651 | 19.2401 | 8.7145 | 5.1121 | 3.3034 | 2.4367 | 1.9583 | 1.6046 |

environments like uncontaminated environments and different contaminated environments with symmetric, asymmetric variance disturbances and non-normal environment.

## Supporting information

**S1 Table. Data generating R program, Table 1 standard deviation code for N(0,1).**
(DOCX)

**S1 File. Data 2 generating R program, S. D codes for G (2,1).**
(DOCX)

**S2 File. Data 3 generating R program, S.D codes for ARL & SDRL.**
(DOCX)

## Author Contributions

**Conceptualization:** Umair Khalil, Tahira Saeed Khan.

**Formal analysis:** Umair Khalil, Tahira Saeed Khan, Dost Muhammad Khan.

**Investigation:** Tahira Saeed Khan.

**Methodology:** Umair Khalil, Tahira Saeed Khan.

**Software:** Tahira Saeed Khan, Dost Muhammad Khan, Muhammad Hamraz.

**Supervision:** Umair Khalil, Walaa Ahmad Hamdi.

**Validation:** Umair Khalil, Dost Muhammad Khan, Muhammad Hamraz.

**Writing – original draft:** Tahira Saeed Khan.

**Writing – review & editing:** Umair Khalil, Walaa Ahmad Hamdi.

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
