## [Decision Letter · Decision Letter 0]

21 Feb 2023

PONE-D-23-01321A ROBUST CUSUM CONTROL CHART FOR MEDIAN ABSOLUTE DEVIATION BASED ON TRIMMING AND WINSORIZATIONPLOS ONE

Dear Dr. Khalil,

Thank you for submitting your manuscript to PLOS ONE. After careful consideration, we feel that it has merit but does not fully meet PLOS ONE’s publication criteria as it currently stands. Therefore, we invite you to submit a revised version of the manuscript that addresses the points raised during the review process.

We look forward to receiving your revised manuscript.

Kind regards,

Kok Haur Ng, Ph.D.

Academic Editor

PLOS ONE

Journal Requirements:

2. Please note that PLOS ONE has specific guidelines on code sharing for submissions in which author-generated code underpins the findings in the manuscript. In these cases, all author-generated code must be made available without restrictions upon publication of the work. 

Please review our guidelines at https://journals.plos.org/plosone/s/materials-and-software-sharing#loc-sharing-code and ensure that your code is shared in a way that follows best practice and facilitates reproducibility and reuse.

3. We noted in your submission details that a portion of your manuscript may have been presented or published elsewhere:

"this paper is written from MPhil thesis of the author Ms. Tahira Saeed Khan who has submitted it for the award of Mphil degree from Abdul Wali Khan University Mardan Pakistan the information as given bellow 

(A ROBUST CUSUM CONTROL CHART FOR MEDIAN ABSOLUTE DEVIATION BASED ON TRIMMED WINSORIZATION by Tahira Saeed Khan

From 2021 (MS/MPHIL Repository (AWK University Mardan))

Processed on 29-Mar-2021 14:33 PKT

ID: 1545291488)"

Additional Editor Comments:

We have completed the review of your manuscript. When revising your manuscript, please consider all issues mentioned in the reviewers' comments, outline every change made in response to their comments and provide suitable rebuttals for any comments not addressed.

Reviewers' comments:

Reviewer's Responses to Questions

**Comments to the Author**

1. Is the manuscript technically sound, and do the data support the conclusions?

Reviewer #1: Yes

Reviewer #2: Yes

2. Has the statistical analysis been performed appropriately and rigorously? 

Reviewer #1: Yes

Reviewer #2: Yes

3. Have the authors made all data underlying the findings in their manuscript fully available?

Reviewer #1: Yes

Reviewer #2: Yes

4. Is the manuscript presented in an intelligible fashion and written in standard English?

Reviewer #1: No

Reviewer #2: No

5. Review Comments to the Author

Reviewer #1: This paper propose a robust CUSUM chart to evaluate the performance of dispersion parameters. The design structure features of various control charts, based on currently defined estimators and some new robust scale estimators by using trimming and winsorization in different scenarios, are examined. The Median Absolute Deviation based on trimming and winsorization is introduced. The performance of the new proposed chart are evaluated in terms of ARL and SDRL.

Please see the attachment for detail review comments.

Reviewer #2: The authors of this paper considers several robust estimators for the CUSUM chart. The performance measures of ARL and SDRL are used to evaluate the performance of the proposed estimators. The paper contains some merit, however, this paper is not well written and cannot be accepted for publication in its present form. This paper needs a major revision and its English must be significantly improved especially for Sections 3, 4 and 5. Details, please refer to the attached referee's report.

6. PLOS authors have the option to publish the peer review history of their article (what does this mean?). If published, this will include your full peer review and any attached files.

Reviewer #1: No

Reviewer #2: No

---

## [Author Response · Author response to Decision Letter 0]

2 Jun 2023

Response to Reviewer II

we tried our level best to incorporate all the major/minor revisions which were raised by both reviewers.

---

## [Decision Letter · Decision Letter 1]

14 Jun 2023

PONE-D-23-01321R1A ROBUST CUSUM CONTROL CHART FOR MEDIAN ABSOLUTE DEVIATION BASED ON TRIMMING AND WINSORIZATIONPLOS ONE

Dear Dr. Khalil,

Thank you for submitting your manuscript to PLOS ONE. After careful consideration, we feel that it has merit but does not fully meet PLOS ONE’s publication criteria as it currently stands. Therefore, we invite you to submit a revised version of the manuscript that addresses the points raised during the review process.

We look forward to receiving your revised manuscript.

Kind regards,

Kok Haur Ng, Ph.D.

Academic Editor

PLOS ONE

Journal Requirements:

Additional Editor Comments:

We have completed the review of the revised manuscript. The authors are required to address all comments given by the reviewer. When editing your manuscript, please consider all issues mentioned in the reviewer' comments, outline every change made in response to the comments and provide suitable rebuttals for any comments not addressed.

Reviewers' comments:

Reviewer's Responses to Questions

**Comments to the Author**

1. If the authors have adequately addressed your comments raised in a previous round of review and you feel that this manuscript is now acceptable for publication, you may indicate that here to bypass the “Comments to the Author” section, enter your conflict of interest statement in the “Confidential to Editor” section, and submit your "Accept" recommendation.

Reviewer #2: (No Response)

2. Is the manuscript technically sound, and do the data support the conclusions?

Reviewer #2: (No Response)

3. Has the statistical analysis been performed appropriately and rigorously? 

Reviewer #2: (No Response)

4. Have the authors made all data underlying the findings in their manuscript fully available?

Reviewer #2: (No Response)

5. Is the manuscript presented in an intelligible fashion and written in standard English?

Reviewer #2: (No Response)

6. Review Comments to the Author

Reviewer #2: The authors of this paper have addressed most of the comments raised in my previous report. However, there are still issues that must be addressed before the manuscript can be accepted for publication. The following are the issues that the authors must address from the manuscript (without track). Please refer to the review report for details.

7. PLOS authors have the option to publish the peer review history of their article (what does this mean?). If published, this will include your full peer review and any attached files.

Reviewer #2: No

---

## [Author Response · Author response to Decision Letter 1]

7 Jul 2023

All the suggestions has been incorporated in the manuscript

---

## [Editor Report · Decision Letter 2]

26 Jul 2023

PONE-D-23-01321R2A ROBUST CUSUM CONTROL CHART FOR MEDIAN ABSOLUTE DEVIATION BASED ON TRIMMING AND WINSORIZATIONPLOS ONE

Dear Dr. Khalil,

Thank you for submitting your manuscript to PLOS ONE. After careful consideration, we feel that it has merit but does not fully meet PLOS ONE’s publication criteria as it currently stands. Therefore, we invite you to submit a revised version of the manuscript that addresses the points raised during the review process.

We look forward to receiving your revised manuscript.

Kind regards,

Kok Haur Ng, Ph.D.

Academic Editor

PLOS ONE

Journal Requirements:

Additional Editor Comments (if provided):

There are some minor comments to improve the manuscript's readability.

Comments:

1. Section 1: Instead of discussing the reviews one by one in different paragraphs, the authors should combine those reviews into the same paragraph.

2. All mathematical notations should be consistent. For example, i) Section 3, lines 2 and 3: What is the difference between italic and nonitalic “Y”? ii) Section 3: What is “C Normal”?, iii) All mathematical symbols should be written using equation editor. For example, n=5. Similar comments applied to other sections.

3. Section 4, line 3: Spacing between theta_hat and chart.

4. Incomplete sentences: i) Section 4: “Table 3 and Table 4… [40]”, ii) Section 4: “The results of CUSUM… [31]”, iii) Section 5.1: The ARL1… charts work well,”. Similar comments applied to other sections.

5. Some special names do not need to capitalize it. For example, i) “Average Run Length (ARL)”-> “average run length (ARL)”, ii) “Modified Trimmed Standard Deviation (MTSD)” -> “modified trimmed standard deviation (MTSD)”, iii) “( IQR)” -> non italic “(IQR)” and etc.

6. The authors should use a consistent font, italic and nonitalic text.

7. All abbreviations should be defined for the first time appear in the text. For example, EWMA.

8. Based on the above comments, the authors require to check the manuscript thoroughly.

---

## [Author Response · Author response to Decision Letter 2]

23 Aug 2023

Dear Reviewer and Academic Editor

we have uploaded a rebuttal letter that responds to each point raised by the academic editor and reviewer with as a separate file labeled 'Response to Reviewers'.

we also have uploaded a marked-up copy of our manuscript that highlights changes made to the 2nd revise version. which has now been labeled as 3rd with track changes": .

An unmarked version of our revised paper without tracked changes has been uploaded as well. we have upload this as a separate file labeled '3rd copy without track changes'.

---

## [Editor Report · Decision Letter 3]

6 Sep 2023

PONE-D-23-01321R3A ROBUST CUSUM CONTROL CHART FOR MEDIAN ABSOLUTE DEVIATION BASED ON TRIMMING AND WINSORIZATIONPLOS ONE

Dear Dr. Khalil,

Thank you for submitting your manuscript to PLOS ONE. After careful consideration, we feel that it has merit but does not fully meet PLOS ONE’s publication criteria as it currently stands. Therefore, we invite you to submit a revised version of the manuscript that addresses the points raised during the review process.

ACADEMIC EDITOR:The authors are required to address all comments accordingly.   

We look forward to receiving your revised manuscript.

Kind regards,

Kok Haur Ng, Ph.D.

Academic Editor

PLOS ONE

Journal Requirements:

Additional Editor Comments:

There are some minor comments to improve the manuscript's readability.

Comments:

1. Section 1: Instead of discussing the reviews one by one in different paragraphs, the authors should combine those similar topics into the same paragraph.

2. Incomplete sentences: i) Section 1: “… qualitatively similar The rest…”, ii) Section 3: “…contamination with sample sizes n=5,6 and. For Gamma distribution…”. Similar comments applied to other sections.

3. Based on the above comments, the authors require to check the manuscript thoroughly.

---

## [Author Response · Author response to Decision Letter 3]

23 Oct 2023

Dear Reviewer,

We greatly appreciate your thoughtful review of our manuscript and your valuable feedback. We have carefully considered your suggestion to combine similar topics into a single paragraph. We recognize the importance of clarity and conciseness in scientific writing and have endeavored to address this issue.

In response to your feedback, we have restructured the manuscript to merge related topics into cohesive paragraphs. This reorganization has allowed us to streamline the presentation of our research findings and eliminate redundancy. Furthermore, we have removed some references that were similar in nature to avoid unnecessary repetition, as per your recommendation.

However, it is worth noting that some of the references (topics) you mentioned are integral to the manuscript, and their inclusion is crucial for providing a comprehensive and accurate account of our research. We believe that these references contribute significantly to the contextual background and overall understanding of our work.

We hope that our revisions align with your expectations and improve the overall readability and coherence of the manuscript. If you have any further suggestions or specific concerns regarding the revised sections or references, please do not hesitate to let us know. Your expertise and feedback are invaluable to us, and we are committed to addressing any remaining issues to enhance the quality of our work.

we have also inserted the tables in the main manuscript

Once again, thank you for your time and effort in reviewing our manuscript.

---

## [Editor Report · Decision Letter 4]

5 Nov 2023

PONE-D-23-01321R4A ROBUST CUSUM CONTROL CHART FOR MEDIAN ABSOLUTE DEVIATION BASED ON TRIMMING AND WINSORIZATIONPLOS ONE

Dear Dr. Khalil,

Thank you for submitting your manuscript to PLOS ONE. After careful consideration, we feel that it has merit but does not fully meet PLOS ONE’s publication criteria as it currently stands. Therefore, we invite you to submit a revised version of the manuscript that addresses the points raised during the review process.

ACADEMIC EDITOR:

There are some minor comments to improve the manuscript's readability.

Comments:

Section 1: The authors rearrange this section of similar topics into several paragraphs.All abbreviations should be defined for the first time appear in the text. For example, CUSUM, SPRT.All names should use consistent font type. For example, “CUSUM” and “*CUSUM*”.There are some grammatical errors. For example, "Table3 and Table 4 shows" and "From Table 7 and Table 8 results of ARL and SDRL shows".Some words and symbols are joining together. For example, "According to Tuprah and Ncube [36] whereY_0=0and the…".Some sentences are incomplete due to missing ".". s

Based on the above comments, the authors are required to check the manuscript thoroughly.

We look forward to receiving your revised manuscript.

Kind regards,

Kok Haur Ng, Ph.D.

Academic Editor

PLOS ONE

Journal Requirements:

Additional Editor Comments (if provided):

There are some minor comments to improve the manuscript's readability.

Comments:

1. Section 1: The authors rearrange this section of similar topics into several paragraphs.

2. All abbreviations should be defined for the first time appear in the text. For example, CUSUM, SPRT.

3. All names should use consistent font type. For example, “CUSUM” and “CUSUM”.

4. There are some grammatical errors. For example, "Table3 and Table 4 shows" and "From Table 7 and Table 8 results of ARL and SDRL shows".

5. Some words and symbols are joining together. For example, "According to Tuprah and Ncube [36] whereY_0=0and the…".

6. Some sentences are incomplete due to missing ".". s

7. Based on the above comments, the authors are required to check the manuscript thoroughly.
---

## [Author Response · Author response to Decision Letter 4]

1 Jan 2024

all the suggestions has been incorporated in the 5th revised version of the manuscri[pt

---

## [Editor Report · Decision Letter 5]

9 Jan 2024

A ROBUST CUSUM CONTROL CHART FOR MEDIAN ABSOLUTE DEVIATION BASED ON TRIMMING AND WINSORIZATION

PONE-D-23-01321R5

Dear Dr. Khalil,

We’re pleased to inform you that your manuscript has been judged scientifically suitable for publication and will be formally accepted for publication once it meets all outstanding technical requirements.

Kind regards,

Kok Haur Ng, Ph.D.

Academic Editor

PLOS ONE

Additional Editor Comments (optional):

The authors have made improvements as suggested. The revised manuscript sounds better.